# A general framework for nitrogen deposition effects on soil respiration in global forests

Xiaoyu Cen [1,2,3,4], Peter Vitousek [3], Nianpeng He [1,2,5] ✉,
Ben Bond-Lamberty [6], Shuli Niu [2], Enzai Du [7], Kailiang Yu [2,8],
Mianhai Zheng[9], Kevin Van Sundert [10], Elizabeth L. Paulus[3,11], Liyin He [12],
Li Xu[2], Mingxu Li [2] & Klaus Butterbach-Bahl [4,13]

Since the Industrial Revolution, human activities have altered atmospheric nitrogen (N) deposition to global forests, affecting carbon dioxide emissions from soils (soil respiration or SR) – one of the largest land-atmosphere carbon fluxes. However, experimental studies have demonstrated both positive and negative effects of N deposition on SR in global forests, leading to debates on how N deposition increases or decreases SR. We developed a framework for generalizing SR responses to N deposition using synthesized data from 168 N addition experiments worldwide and observed SR across the global natural N deposition gradient. The findings indicate that N deposition decreased SR in 2.9% of global forested areas, particularly in eastern China, western Europe, and the eastern USA. However, the net effect of N deposition increased the global forest SR by ~5% ($1.7 \pm 0.1$ PgC yr$^{-1}$). If N pollution could be effectively controlled, global forest SR would decrease, potentially contributing to a reduction in the terrestrial carbon emissions.

Globally, carbon dioxide ($CO_2$) emission from soils (soil respiration or SR; Box 1) is one of the largest carbon fluxes between the land and atmosphere, estimated to be 7–8 times greater than current anthropogenic $CO_2$ emissions[1–3]. Forests cover about one-third of the world's land area, and their SR rates are higher than those of other ecosystems[4], making forest soils an important component of the global carbon cycle. Due to the high carbon sequestration potential of forest ecosystems, afforestation, reforestation and sustainable forest management have become the preferred natural climate solutions in many countries aiming to achieve net zero emissions (or carbon neutrality) by the mid-21st century[5–7]. Accordingly, evaluating the effects of human-induced disturbances on forest SR is essential for assessing changes in the global carbon cycle and for optimizing climate change mitigation efforts.

Human activities, such as fertilizer use, fossil fuel combustion, vehicle emissions, etc., have released large amounts of reactive nitrogen (N) into the atmosphere. As a result, the annual flux of N deposited to terrestrial ecosystems has more than tripled globally compared to

[1]Key Laboratory of Boreal Forest Ecosystem Conservation and Restoration, National Forestry and Grassland Administration, Harbin, China. [2]Key Laboratory of Ecosystem Network Observation and Modeling, Institute of Geographic Sciences and Natural Resources Research, Chinese Academy of Sciences, Beijing, China. [3]Department of Earth System Science, Stanford University, Stanford, CA, USA. [4]Pioneer Center Land-CRAFT, Department of Agroecology, Aarhus University, Aarhus C, Denmark. [5]Institute of Carbon Neutrality, School of Ecology, Northeast Forestry University, Harbin, China. [6]Joint Global Change Research Institute, Pacific Northwest National Laboratory, College Park, MD, USA. [7]State Key Laboratory of Earth Surface Processes and Disaster Risk Reduction, Faculty of Geographical Science, Beijing Normal University, Beijing, China. [8]High Meadows Environmental Institute, Princeton University, Princeton, NJ, USA. [9]Key Laboratory of Vegetation Restoration and Management of Degraded Ecosystems, Guangdong Provincial Key Laboratory of Applied Botany, South China Botanical Garden, Chinese Academy of Sciences, Guangzhou, China. [10]Department of System Earth Science, Maastricht University, Innovalaan 1, Venlo, the Netherlands. [11]Geochemistry and Biogeochemistry Group, SLAC National Accelerator Laboratory, Menlo Park, CA, USA. [12]Nicholas School of the Environment, Duke University, Durham, NC, USA. [13]Institute for Meteorology and Climate Research, Atmospheric Environmental Research, Karlsruhe Institute of Technology, Garmisch-Partenkirchen, Germany. ✉e-mail: henp@igsnrr.ac.cn

# BOX 1
# Glossary of terms and abbreviations

Nitrogen availability: Nitrogen supply in relative to biotic nitrogen demand (sensu Mason et al.[84]). Increased nitrogen availability (e.g., due to high anthropogenic nitrogen deposition) can cause nitrogen-limited ecosystems to become nitrogen-saturated.

Nitrogen status: The state of an ecosystem with respect to its nitrogen availability. In this study, we used nitrogen limitation/saturation status to distinguish between two types of forests with different nitrogen availability that exhibit disparate responses to nitrogen inputs.

Probability of soil respiration being decreased by nitrogen input ($P_{dec}$): Depending on the background nitrogen availability and the nitrogen tolerance level of organisms in a forest ecosystem, additional nitrogen inputs could increase or decrease soil respiration. The probability of soil respiration being decreased by nitrogen input was statistically inferred from the frequency with which a nitrogen input rate was observed to reduce soil respiration in experimental forests.

Response factor: A quantitative metric for measuring the effect size of nitrogen input on soil respiration. Considering the mechanistic differences, a positive response factor ($f_{pos}$; calculated as the increased soil respiration per unit of added nitrogen) was used to measure the increased soil respiration by nitrogen inputs, which is likely dominated by the continuous, nitrogen-facilitated biomass growth. A negative response factor ($f_{neg}$; calculated as the decreased soil respiration relative to the initial soil respiration) was used to measure the decreased soil respiration by nitrogen inputs, which is likely dominated by the abrupt, N-induced species loss.

Soil respiration (SR): Carbon dioxide emissions from the soil, the majority of which come from microbial respiration (MR) and plant root respiration (RR).

preindustrial levels[8,9]. Despite recent declines in N deposition in some regions[9,10], high anthropogenic N deposition in past decades may have caused many forests to become N-saturated (i.e., N supply exceeds biotic N demand)[11], particularly tropical and temperate forests close to human settlements[12,13]. Meanwhile, N limitation is widespread in forests far from human activities, in regenerating young forests, and in boreal forests[14]. In N-saturated versus N-limited forests, plant and microbial processes (such as net primary production and soil N mineralization) may respond differently to N deposition[11,15], potentially leading to varying effects of N deposition on SR—the sum of plant root respiration and microbial respiration.

Over the past half century, N addition experiments conducted in forests worldwide have reported both positive and negative responses of SR to N inputs[16–18]. Meta-analyses of N addition experiments have demonstrated the important role of N input rate and forest background N status (Box 1) on SR responses[19–22]. However, the analyses reached divergent conclusions as to whether N deposition increased or decreased the gross SR budget in global forests[19–22]. Early meta-analysis using experimental N addition data revealed that N deposition reduced SR in global forests especially in N-rich forests[19,23], whereas some recent meta-analyses using updated data sets have suggested that realistically low N deposition levels increased global forest SR[20,21]. The N deposition-induced reduction of SR is consistent with the N saturation hypothesis[11,24], which predicts the negative responses of SR when forests have high N availability and become N-saturated. On the other hand, there are observations that low N inputs increase SR even in some N-rich forests and in forests receiving long-term N additions[16,25]. Such discrepancy implies the need to reevaluate the N deposition-induced changes in forest SR, in order to provide a general explanation for the diverse effects of N deposition on SR in N-limited and N-saturated forests.

Previous research has shown that N inputs can both facilitate biomass growth and induce species loss[26]. N-facilitated, continuous biomass growth could increase SR. Conversely, decrease in SR may result from N-induced species loss, which often occurs abruptly. The different mechanisms that dominate SR changes (see Fig. 1) imply that increases and decreases in SR caused by N inputs must be analyzed separately. Based on biological and ecological theories, we first proposed a hypothesis that the increased SR by N inputs would follow a biphasic response pattern (Fig. 1)[11,27–29]. Short-term, low N inputs often promote plant and microbial growth[30,31] and exoenzyme synthesis[32,33]. Given the power-law relationship between increased plant and microbial biomass and metabolic respiration[27] and the positive relationship between exoenzyme concentration and soil organic matter

decomposition rate[29], there may be a positive dose-response relationship between low N inputs and increased SR (Supplementary Fig. S1). However, high exogenous N supply, such as from atmospheric deposition, can exceed the ability of plants and microbes to maintain cellular homeostasis and become toxic to organisms[34,35]. Soil acidification caused by excessive N can be detrimental to certain species. Depletion of labile organic carbon in soils may also limit the N-facilitated biomass growth and the positive response of SR. These factors may contribute to a biphasic dose-response relationship between N inputs and the increased SR (i.e., higher increment under low N, and lower increment under high N)[36].

The N toxicity and low soil pH caused by high N inputs could reduce respiration at the organism or population level. But at the community level, biodiversity and functional compensation among species ensure community stability under changes in N availability within a certain range[37]. When high N availability exceeds a critical threshold, however, species adapted to N-poor environments cannot survive[38,39], and community functions—including plant productivity and microbial decomposition—can be suddenly weakened[28]. Thus, soil respiration, which is controlled by plant and microbial activities, may abruptly decrease under excessively high N availability. Such phased transitions of microbial biomass and plant productivity were observed in an experimental forest receiving long-term N additions, where SR decreased significantly during the transition period[40]. Based on these theories and observations, we hypothesized that the decreased SR by N inputs would follow an abrupt transition pattern, as outlined in Fig. 1.

This study was designed to (1) reveal the forest SR increase and decrease patterns under changes in N deposition and experimental N addition, (2) provide a framework for generalizing the N deposition effects on SR in N-limited and N-saturated forests, and (3) assess the net effect of N deposition on the annual SR budget in global forests. In this study, we compiled data from 168 experimental N addition forests worldwide ($CO_2$_exp dataset; Fig. 2). At each site, SR response factors (Box 1) were calculated for each N addition level and year, as metrics to assess the effect sizes of N inputs (see Supplementary Text S1 for the response factors, their theoretical basis and calculation). Quantitative models of the response factors were constructed to validate the hypothesized positive and negative response patterns of SR (Fig. 2), and also to estimate the altered SR by current N deposition levels. Meanwhile, we used SR observations in non-experimental forests from a published database (SRDB)[41] to check the change in SR along the natural gradient of N deposition, and also to derive the annual SR budget in global forests.

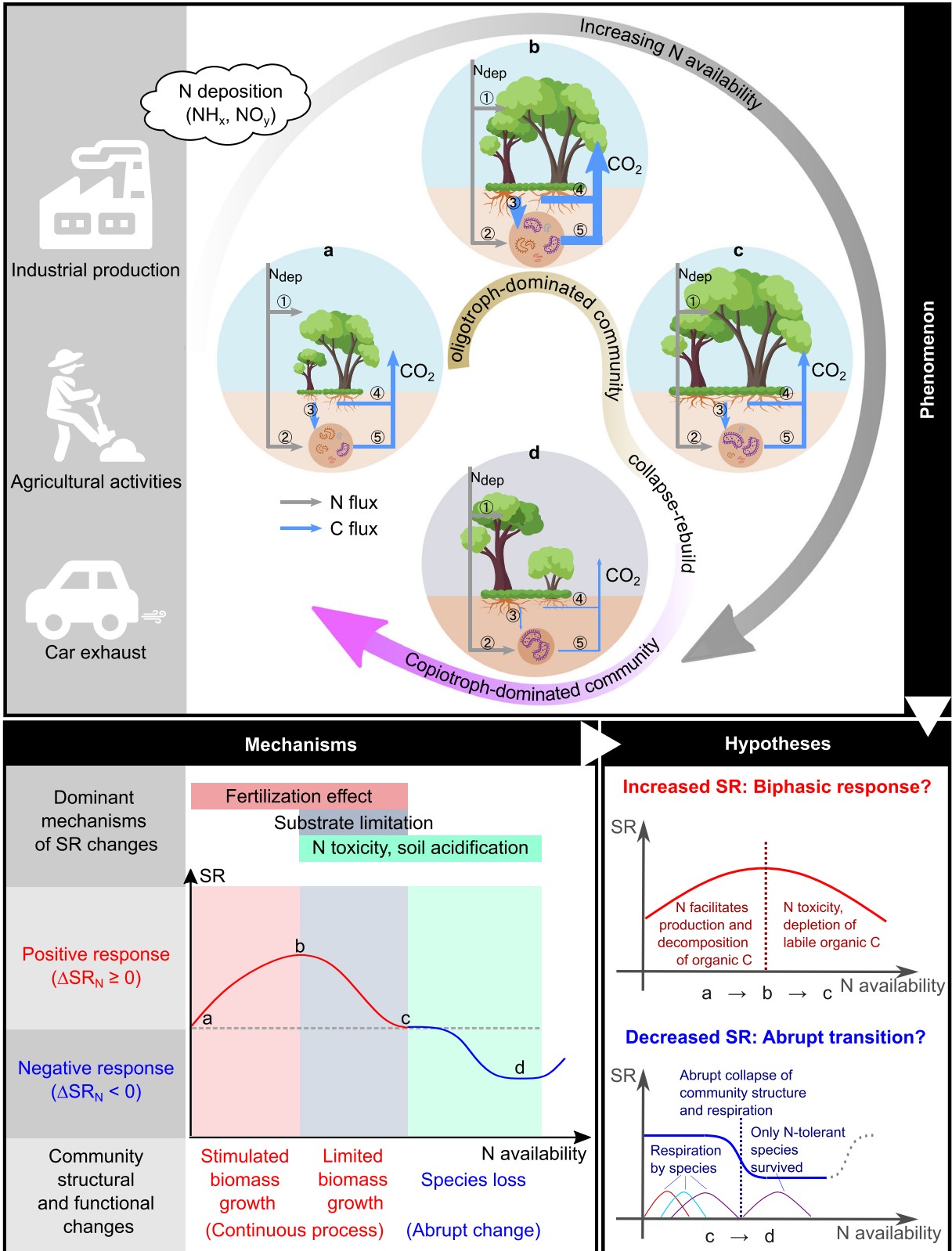

**Fig. 1 | Hypothetical change in soil respiration (SR) under increasing N availability.** Panels (**a**–**d**) show the different stages that a N-limited forest can go through under increasing N availability. Initially, N deposition ($N_{dep}$) increases SR in the N-limited forest, but the positive effect gradually declines due to the buildup of N toxicity and the depletion of substrate. Subsequently, an oligotroph-dominated community in the N-limited forest collapses under excessively high N availability and rebuilds into a copiotroph-dominated community (when the forest becomes N-saturated; **d**). (1) Canopy uptake of deposited N; (2) Microbial assimilation; (3) Root exudate containing labile organic carbon; (4) Root respiration; (5) Microbial heterotrophic respiration. $\Delta SR_N$: N deposition-induced soil respiration changes. This figure was created using free resources from the iSlide platform (www.islide.cc) under the CC0 copyright agreement.

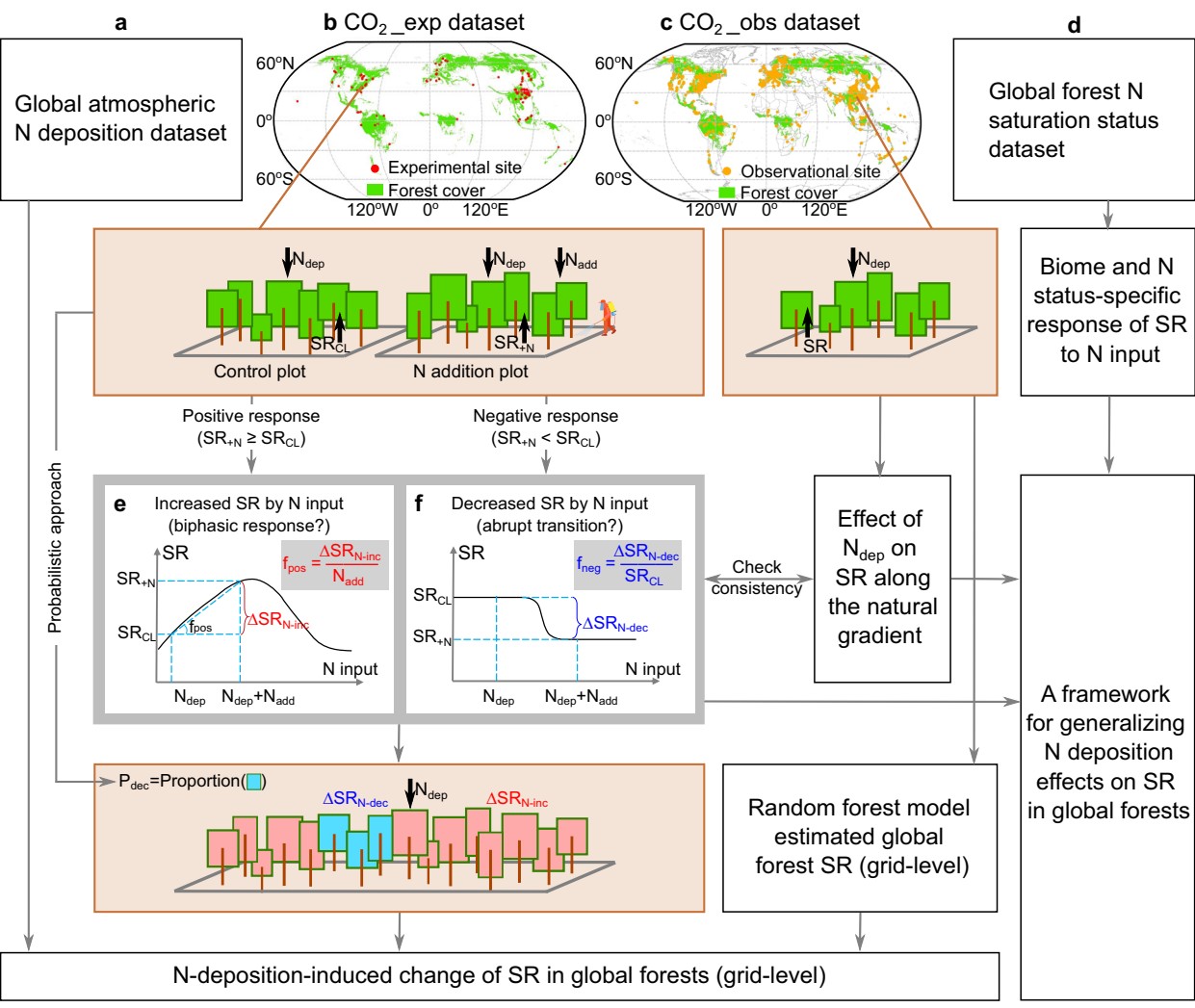

**Fig. 2 | Schematic representation of the data analysis workflow for this study.**
**a** Global atmospheric N deposition data were from Ackerman et al.[9]. **b** Experimental N addition forests with observations of soil respiration ($CO_2$_exp dataset). **c** Non-experimental forests with observations of soil respiration ($CO_2$_obs dataset)[41]. Basemap was from Resource and Environmental Science Data Platform (www.resdc.cn). **d** Global forest N saturation status data were from Cen et al.[13]. **e** Hypothesized response of the increased soil respiration (SR) to N inputs. **f** Hypothesized response of the decreased SR to N inputs. $SR_{CL}$: soil respiration rate measured in the control plot receiving natural nitrogen deposition ($N_{dep}$). $SR_{+N}$: soil respiration rate measured in the experimental plot receiving an additional N input ($N_{add}$). $\Delta SR_{N-inc}$: increased SR by N input. $\Delta SR_{N-dec}$: decreased SR by N input. $f_{pos}$: positive response factor of increased SR to N input. $f_{neg}$: negative response factor of decreased SR to N input (see Supplementary Text S1 for the biochemical and ecological theories underlying the response factors). $P_{dec}$: Probability of SR being decreased by N input, estimated as the proportion of experimental N addition forests where a N input rate caused a decreased in soil respiration.

## Results

### Increased and decreased soil respiration by N inputs

Due to the different dominant mechanisms underlying the positive and negative responses, which are likely dominated by continuous biomass change and abrupt species loss, respectively, different mathematical models were required to quantify the SR responses (see Figs. 1, 2e, f). Thus, we analyzed the increased and decreased SR by N inputs separately.

Using data from experimental forests where N addition increased SR, we found that across forest biomes and N status, the positive response factor ($f_{pos}$) of SR was negatively related to low N inputs; $f_{pos}$ was barely related to high N inputs (see Supplementary Fig. S2 for regression models and change points; details of piecewise linear regression models were provided in Supplementary Table S1). The negative relationship between $f_{pos}$ and low N inputs could be described using linear regression models (Fig. 3; Supplementary Fig. S3). Such a negative relationship between low N inputs and $f_{pos}$ (which is like the sensitivity of increased SR to N inputs) suggests an inverted-U-shaped

quadratic relationship between N inputs and increased SR, supporting the hypothesized biphasic response of increased SR to N inputs (Fig. 2e). In the Methods section, we provided an equational inference of how the relationship between N inputs and $f_{pos}$ translated into the quadratic relationship between N inputs and increased SR (see Eq. 3). Additionally, the quadratic relationship was validated at the site scale using data from four experimental sites with at least five levels of N inputs (Supplementary Fig. S4).

Random forest models built with the experimental dataset ($CO_2$_exp) showed that N input rate was one of the most important factors in explaining the negative SR responses (as measured by the negative response factor $f_{neg}$; see Supplementary Table S2 for model parameters). However, $f_{neg}$ did not change gradually with increasing N input as $f_{pos}$ did. Statistically, no reduction in SR was observed in N addition experiments when N input rates were not higher than 15 kgN ha$^{-1}$ yr$^{-1}$ (Supplementary Table S3). When N input rates were between 15–100 kgN ha$^{-1}$ yr$^{-1}$, mean $f_{neg}$ of experimental forests remained nearly constant, regardless of the forest biome or N status (Fig. 4;

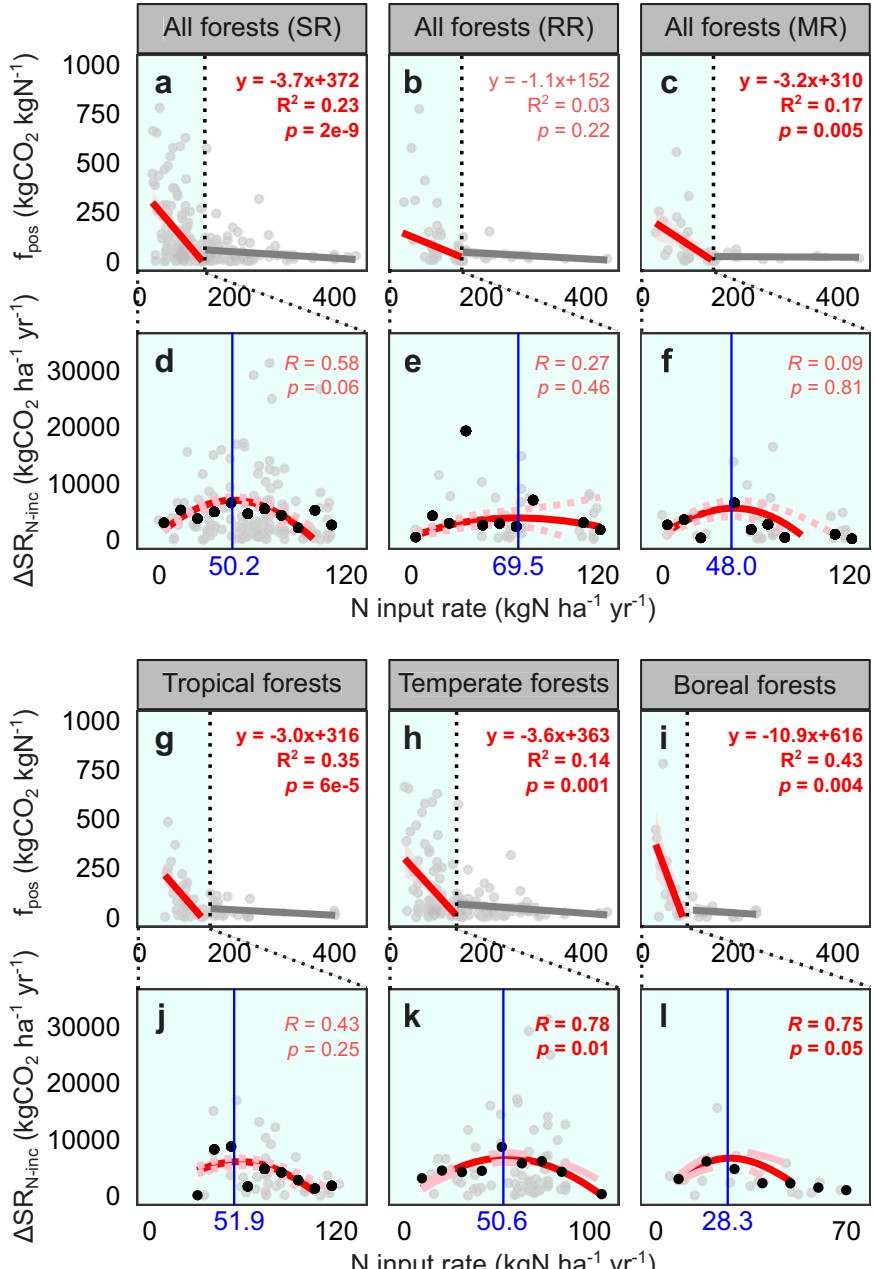

**Fig. 3 | Positive effects of experimental N input on soil respiration (SR) and its components, root respiration (RR) and microbial respiration (MR).**
**a–c**, **g–i** Piecewise linear regression models relating N input rate to the response factor of increased SR to N input ($f_{pos}$) in experimental forests. Each gray point represents a $f_{pos}$ calculated using data from a pair of control and N addition plots from the same site-year (CO$_2$_exp). Red (gray) lines show the fitted linear models for $f_{pos}$ and low (high) N input rate. Red text shows statistically significant (i.e., $p < 0.05$) models; pink text shows insignificant models. The linear models for $f_{pos}$ and low N input rate (red text) were then used to infer the quadratic models of increased SR ($\Delta SR_{N\text{-}inc}$) and N input. **d–f**, **j–l** Quadratic models relating the increased soil respiration ($\Delta SR_{N\text{-}inc}$) to N input rate in forests. Each gray point represents a $\Delta SR_{N\text{-}inc}$, calculated as the difference in SR of a paired control and N addition plots from the same site-year. Black points show the moving average of $\Delta SR_{N\text{-}inc}$ for every 10 kgN ha$^{-1}$ yr$^{-1}$; this aggregation of data was done to reduce the influence of extreme values. Red lines show the quadratic models. Pink dashed lines show the range of uncertainties of the quadratic models. Red text shows statistically significant correlation coefficient ($R$) between the observed and predicted mean $\Delta SR_{N\text{-}inc}$; pink text shows insignificant $R$. Blue lines and text show the turning points of the quadratic models.

Supplementary Fig. S5). When the N inputs were further increased to between 100–200 kgN ha$^{-1}$ yr$^{-1}$, $f_{neg}$ values suddenly dropped, indicating a pronounced decrease in SR. Under N inputs of 200–400 kgN ha$^{-1}$ yr$^{-1}$, $f_{neg}$ stabilized again. The results supported our hypothesis that the decreased SR by N input followed an abrupt transition pattern.

Also, we analyzed the N effects on the two components of SR, root respiration and microbial respiration. The N-induced increases in root respiration and microbial respiration followed inverted-U-shaped patterns. The N input rate corresponding to the peak increase in microbial respiration was lower than that corresponding to the peak increase in root respiration (Fig. 3e, f). Abrupt transitions were found in the decreased root respiration and microbial respiration by N input. The magnitude of the reduction in root respiration was larger than that in microbial respiration (as indicated by the $f_{neg}$ values). The abrupt transition in the response of microbial respiration occurred at a lower

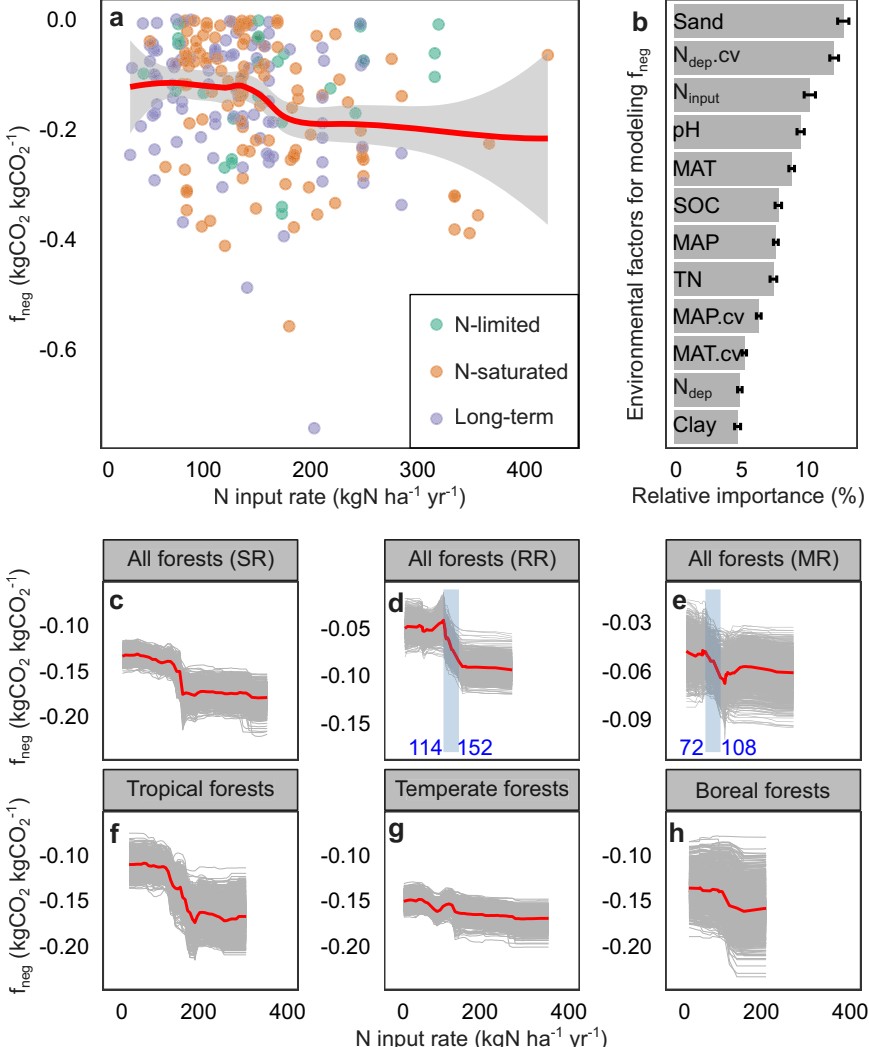

**Fig. 4 | Negative effects of experimental N input on soil respiration (SR) and its components, root respiration (RR) and microbial respiration (MR). a** Local polynomial regression (LOESS) model on N input rate and the negative response factor ($f_{neg}$) of decreased SR to N input. Points in different colors represent data from N-limited, N-saturated, and long-term (i.e., N applied for above 3 years) experimental forests. The red line indicates the LOESS model, and the gray shading indicates the standard error of the model. **b** The relative importance of N input rate and environmental factors for $f_{neg}$. Error bars show the standard errors of the relative importance values. MAT: mean annual temperature; MAP: mean annual precipitation; $N_{dep}$: mean annual N deposition; Sand: soil sand content; Clay: soil clay content; MAT.cv, MAP.cv and $N_{dep}$.cv are the corresponding coefficients of temporal variation; pH: soil pH; SOC: soil organic carbon content; TN: soil total nitrogen content; $N_{input}$: N input rates in the experimental plots with artificial N addition. **c–h** Partial dependence of $f_{neg}$ on N input in experimental forests. In each panel, gray curves show partial dependence plots derived from 1000 random forest models built using 1000 different subsets of the experimental data (to avoid the derived pattern being driven by a few observations), and the red curve shows the arithmetic mean of the 1000 gray curves. Blue shading and text show the turning points in the responses of decreased RR (or MR) to N inputs.

N input rate than the abrupt transition in the response of root respiration (Fig. 4d, e).

An additional analysis of the integrated response of SR to N inputs (i.e., without separating the positive and negative responses) provides qualitative support for the biphasic and abrupt transition patterns derived from our separate analyses of the positive and negative responses. These analyses revealed the same patterns: N inputs first increase, then decrease, and then increase SR in N-limited forests (see Supplementary Text S2 and Fig. S6 for details).

**Effect of natural N deposition on soil respiration in forests**
Using SR observations from non-experimental forests ($CO_2$_obs), we built random forest models to quantify the partial dependence of SR on natural N deposition rates (see Supplementary Table S4 for the model parameters). This analysis statistically removed the influence of factors other than N deposition (e.g., climatic and edaphic factors).

Data from N-limited and N-saturated forests were separated based on the global forest N status map[13].

Across all forest biomes and N status types, inverted-U shaped response patterns were detected in the N deposition-induced SR changes (Fig. 5). This finding was consistent with the responses revealed by experimental data (Figs. 3, 4; Supplementary Fig. S4). Under high N deposition, SR decreased. The declining trends were more pronounced in tropical forests and N-saturated forests than in other forests, suggesting that N deposition is more likely to reduce SR where background N availability is high.

**Probability of soil respiration being decreased by N input**
In the compiled experimental dataset ($CO_2$_exp), both increased and decreased SR were observed even at the same N input rate. Based on the proportions of experimental forests where SR was decreased, we estimated the probability of SR being decreased ($P_{dec}$) by N input in

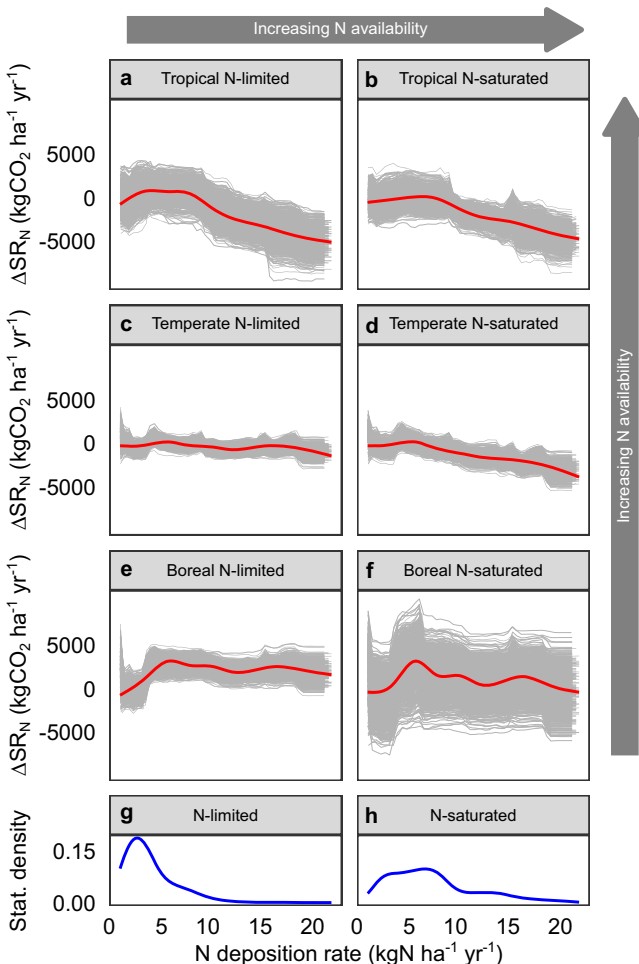

**Fig. 5 | Effect of natural N deposition on observed soil respiration (SR) in forests.** To remove the influence of basal SR differences across biomes and N status, N deposition-induced SR change ($\Delta SR_N$) was estimated as the difference in SR between forests with N deposition rates below 1 kgN ha$^{-1}$ yr$^{-1}$ and forests with N deposition rates above the level. Panels (**a**–**f**) are the partial dependence plots showing the change in soil respiration ($\Delta SR_N$) along the natural gradient of N deposition in different forest biome-N status combinations. These plots were derived from random forest regression models fitted with observational data (CO$_2$_obs dataset). Gray curves in each panel show 1000 partial dependence curves derived from 1000 random forest models built with 1000 different subsets of CO$_2$_obs corresponding to a forest biome-N status combination (this was done to remove the influence of random sampling on the revealed pattern). The red curves show the mean partial dependence of $\Delta SR_N$ on N deposition. Panels (**g**, **h**) are density curves showing the statistical distributions of N deposition rates in global N-limited and N-saturated forests.

forests. Considering the potential influence of background N status and duration of N addition on the responses of SR, we separately analyzed experimental data from initially N-limited forests where N addition experiments were conducted for no more than 3 years (N-limited forests), data from initially N-saturated forests where N addition experiments were conducted for no more than 3 years (N-saturated forests), and data from forests where N addition experiments were conducted for more than 3 years (Long-term experimental forests). Data from long-term experimental forests were treated separately because the accumulated artificial N additions could completely override the background, natural N status of the forests.

Generally, $P_{dec}$ increased with increasing N availability. For any given N input rate, $P_{dec}$ was the lowest in N-limited forests, the second lowest in N-saturated forests, and the highest in long-term experimental forests (Fig. 6). Within a given forest type, $P_{dec}$ was positively correlated with N input rate. Using nonlinear regression models, we estimated the $P_{dec}$ of global forests using the current level of N deposition as N input data. The area-weighted global mean of $P_{dec}$ was 2.9%, meaning that the current N deposition decreased SR in 2.9% of global forests (Supplementary Table S5). $P_{dec}$ was estimated to be high in temperate forests of Eastern China, Western Europe, Eastern United States, and some tropical forests. In short-term experiments ($\leq 3$ years) with N addition rates no higher than 100 kgN ha$^{-1}$ yr$^{-1}$, sites observing decreased SR were mostly in Eastern China, Western Europe, and Eastern United States, consistent with the spatial pattern of the estimated $P_{dec}$ (Fig. 6e).

### Contribution of N deposition to global forest soil respiration

We built random forest regression models using SR observed under natural conditions (CO$_2$_obs dataset; see Supplementary Table S4 for the model parameters) and estimated the SR in global forests at the grid scale (Fig. 7). The correlation coefficient between observed and predicted SR was 0.71 ($p < 2e-16$). We found that the annual SR of global forests was 34.4 PgC yr$^{-1}$. As expected, annual SR generally decreased from low to high latitude forests.

We modeled the response factors ($f_{pos}$ and $f_{neg}$) corresponding to the current level of N deposition (Supplementary Table S5), and then calculated the change in SR due to N deposition in global forests. In each forest grid (0.5°×0.5°, which contains many forests), we estimated the proportion of forests where SR was decreased by N deposition (equal to $P_{dec}$ based on the Law of Large Numbers) and derived the grid-level net change in SR caused by the current level of N deposition. We found that hotspots of changed SR due to N deposition were mostly located in Eastern China, Western Europe, and the Eastern United States (Fig. 7).

Globally, we found that current levels of N deposition increased SR in most forests, by a total of $1.8 \pm 0.1$ PgC yr$^{-1}$. Meanwhile, N deposition decreased SR in other forests, with a total reduction of $0.1 \pm 0.01$ PgC yr$^{-1}$. The net effect was that N deposition increased annual SR by 4.9% ($+1.7$ PgC yr$^{-1}$; i.e., a net effect size of $+84$ kgC kgN$^{-1}$) in global forests (Table 1; Supplementary Table S5).

## Discussion
### A general framework for the effects of N deposition on soil respiration

Combining the general patterns of how N inputs increase and decrease SR (Figs. 3–6), we developed a universal framework (Fig. 8) that could potentially explain the diverse effects of N deposition on SR in N-limited and N-saturated forests.

Essentially, the response curve (Fig. 8) is shaped by the biphasic effect of N input on biotic respiration and the different N tolerance levels of organisms. Biological studies have shown biphasic responses of species to increasing N availability[34,42]. Low N stimulates biomass growth and respiration, but the stimulatory effect diminishes or even becomes inhibitory at high N levels[24]. The differentiated, biphasic responses of species respiration to a range of N inputs (niche complementarity; Fig. 8) add up to be a community-level biphasic response of SR[43,44]. Niche complementarity allowed the stress from N availability changes to be shared among species and sustained the functional stability of the community, but excessively high N availability could exceed the N tolerance limit of the entire community and kill the N-sensitive oligotrophs[45], leading to abrupt reductions in the living biomass and respiration of the community[28].

Such critical transitions in the community structure and respiration have been observed in N addition experiments[40,46]. Furthermore, the continuous transition from an oligotroph- to a copiotroph-dominated community may cause a bimodal response of SR to N input in N-limited forests, due to the different inverted-U-shaped responses of the old and new communities, and the collapse of SR during the transition (Fig. 8). Analysis using experimental data

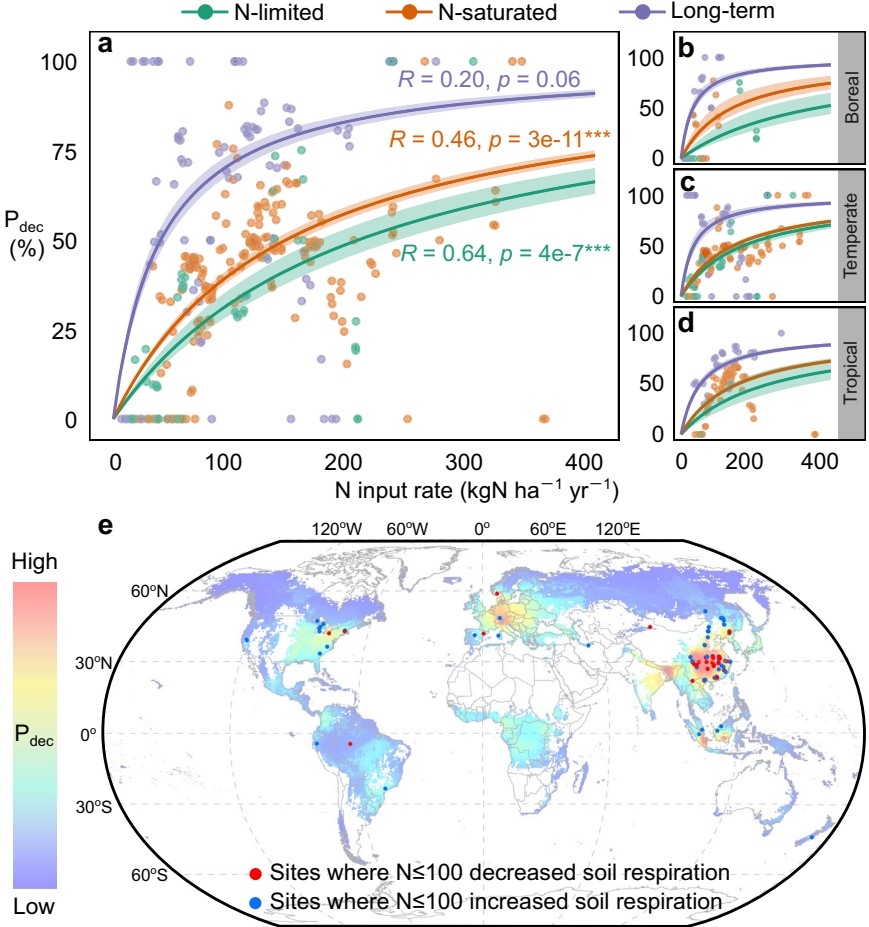

**Fig. 6 | Probability of forest soil respiration being decreased (P_dec) by N input.** Panel (**a**) shows $P_{dec}$ estimated using data from all experimental forests, panels (**b**–**d**) show $P_{dec}$ estimated using data from each biome. In panels (**a**–**d**), each point represents an observed probability (frequency) of soil respiration being decreased by N input corresponding to a given N input rate. Curves and text of the same color show the fitted non-linear models for $P_{dec}$ (***$p < 0.001$). Shadings of the corresponding color show the ranges of uncertainty of the non-linear models. In panel (**e**), the colored map shows the modeled $P_{dec}$ under current levels of atmospheric N deposition. Blue dots indicate the experimental sites where a N addition rate of no more than 100 kgN ha⁻¹ yr⁻¹ (N ≤ 100) increased soil respiration (133 site-year-N addition level combinations); red dots indicate the sites where N ≤ 100 decreased soil respiration (78 site-year-N addition level combinations). Note that long-term experimental forests, where N addition experiments were conducted for more than 3 years, are not shown in panel (**e**). This is because SR in long-term experimental forests could deviate from their natural responses due to the high N availability from accumulated exogenous N inputs. Basemap was from Resource and Environmental Science Data Platform (www.resdc.cn).

from N-limited forests validated the hypothesized bimodal response of SR to N input (see Supplementary Text S2 for details; Fig. S6), supporting the reliability of the universal response curve we proposed.

Both experimental and observational data agreed that low N inputs could increase SR even in N-saturated forests (Fig. 5; Supplementary Fig. S3), probably because the reconstructed copiotroph-dominated community could tolerate and further benefit from the high N availability. Incorporating the community collapse-reconstruction mechanism into the traditional N saturation theory may help explain why low N inputs have been found to increase SR in some N-rich forests and in forests receiving long-term N additions[16,25]. The SR responses of N-saturated forests differ from those of N-limited forests in that for the same level of N input, SR is more likely to decrease in N-saturated than in N-limited forests (Fig. 6). Whether N increases or decreases SR may depend on the background N availability and the community resilience to additional N inputs. Therefore, the relatively high $P_{dec}$ in N-saturated forests may be partly attributable to the initial high N availability. Moreover, the lower biodiversity and community complexity

in N-saturated forests (as compared to N-limited forests)[46,47] may reduce community resilience to additional N inputs, leading to a high risk of community structure change and respiration decline.

We postulate that community collapse-reconstruction (Fig. 1; Fig. 8) may be a general strategy for communities to cope with excessive environmental stress. In a long-term warming experiment, a transient but significant decrease in SR was observed, which was attributed to microbial community reconstruction[48]. This implies that in community- or ecosystem-level studies, researchers should consider non-linear, abrupt changes. Such abrupt transitions challenge accurate predictions of ecosystem dynamics under environmental change. But the predictability of ecosystem functioning can be improved by enlarging the spatial-temporal scale of prediction[49]. For instance, a probabilistic approach was used in this study to predict the changes in SR at the regional scale (see the Methods section for the calculation of $P_{dec}$). From this perspective, data from well-coordinated global experimental networks can be integrated to predict ecosystem functioning in the context of a changing world with ubiquitous abrupt transitions.

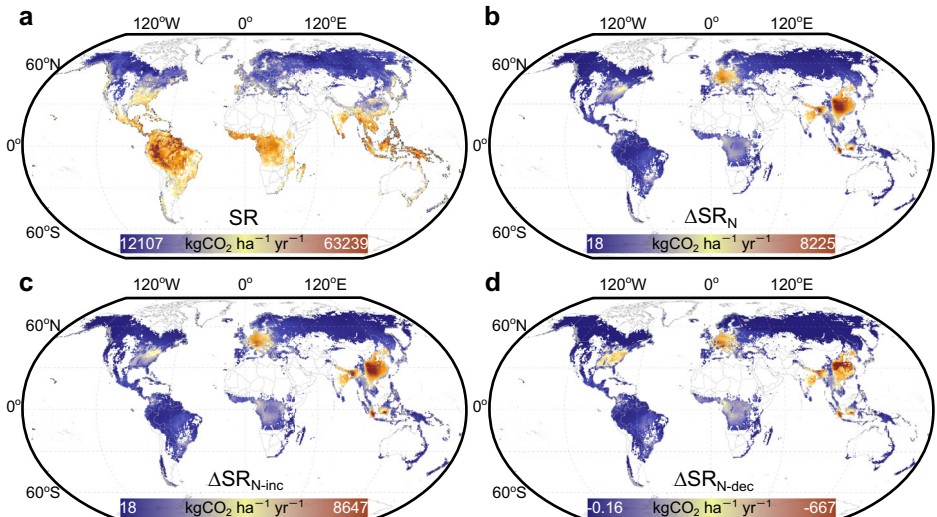

**Fig. 7 | Spatially varying effects of N deposition on forest soil respiration.** Global maps of forest soil respiration rate (SR; **a**), the N deposition-induced change in forest soil respiration rate ($\Delta SR_N$; **b**), the increased soil respiration by N deposition ($\Delta SR_{N\text{-}inc}$; **c**), and decreased soil respiration by N deposition ($\Delta SR_{N\text{-}dec}$; **d**). Each grid contains forests where SR has increased and forests where SR has decreased, so the $\Delta SR_{N\text{-}inc}$ and $\Delta SR_{N\text{-}dec}$ shown at the grid level have been rectified for the proportion of forests where the current N deposition has increased and decreased SR, respectively (see Eq. 10 and Eq. 11 in the Methods section for calculation). For each grid, $\Delta SR_N$ was calculated as the sum of $\Delta SR_{N\text{-}inc}$ and $\Delta SR_{N\text{-}dec}$. Basemap was from Resource and Environmental Science Data Platform (www.resdc.cn).

**Table 1 | Contribution of atmospheric N deposition to global forest soil respiration**

| Biome | N status | Area ($10^8$ ha) | $N_{dep}$ (kgN ha$^{-1}$ yr$^{-1}$) | SR (kgCO$_2$ ha$^{-1}$ yr$^{-1}$) | $\Delta SR_{N\text{-}inc}$ (PgC yr$^{-1}$) | $\Delta SR_{N\text{-}dec}$ (PgC yr$^{-1}$) | $\Delta SR_N$ (PgC yr$^{-1}$) | Annual SR budget (PgC yr$^{-1}$) | Contribution rate (%) |
|---|---|---|---|---|---|---|---|---|---|
| Tropical | N-limited | 8.4 | 3.9 | 42420.1 | 0.2 (0.05) | −0.01 (0.004) | 0.2 (0.05) | 10.1 | 2.1 |
| | N-saturated | 9.6 | 7.2 | 39781.6 | 0.8 (0.1) | −0.05 (0.01) | 0.7 (0.1) | 10.4 | 6.8 |
| | Subtotal | 18.0 | 5.7 | 40907.4 | 1.0 (0.1) | −0.06 (0.01) | 0.9 (0.1) | 20.5 | 4.5 |
| Temperate | N-limited | 3.6 | 5.4 | 28360.2 | 0.1 (0.03) | −0.01 (0.002) | 0.1 (0.03) | 2.8 | 4.5 |
| | N-saturated | 3.8 | 10.5 | 29488.1 | 0.4 (0.06) | −0.03 (0.005) | 0.4 (0.06) | 3.1 | 12.3 |
| | Subtotal | 7.4 | 7.5 | 28842.4 | 0.5 (0.07) | −0.04 (0.005) | 0.5 (0.07) | 5.9 | 8.6 |
| Boreal | N-limited | 10.0 | 2.1 | 21896.6 | 0.2 (0.04) | −0.001 (5e-4) | 0.2 (0.04) | 6.1 | 2.8 |
| | N-saturated | 3.0 | 2.5 | 22820.4 | 0.1 (0.01) | −0.001 (5e-4) | 0.1 (0.01) | 1.9 | 4.6 |
| | Subtotal | 13.0 | 2.2 | 22084.5 | 0.3 (0.04) | −0.002 (7e-4) | 0.3 (0.04) | 8.0 | 3.2 |
| Total | | 38.4 | 4.6 | 29924.9 | 1.8 (0.1) | −0.1 (0.01) | 1.7 (0.1) | 34.4 | 4.9 |

$N_{dep}$: atmospheric N deposition rate; SR: soil respiration rate; $\Delta SR_{N\text{-}inc}$: increased soil respiration by N deposition; $\Delta SR_{N\text{-}dec}$: decreased soil respiration by N deposition; $\Delta SR_N$: net change in soil respiration due to N deposition.
Numbers in parentheses indicate the standard errors of the mean values.

## Responses of plant root respiration and microbial respiration to N inputs

The responses of plant root respiration and microbial respiration to N inputs were consistent with the hypothesized biphasic and abrupt transition patterns (Figs. 3, 4, 8). N fertilization may facilitate plant and microbial growth, thereby enhancing both root respiration and microbial respiration. With increased N availability, however, plants allocate more biomass aboveground and less belowground[50,51], leading to a reduction in root exudate and rhizosphere respiration. Moreover, excessive N can cause the mortality of plants and microbes[18,52], lowering respiration. Previous meta-analyses of global N addition experiments have found that the N-induced changes in SR are positively related to changes in microbial biomass and fine root biomass[53–55], suggesting that N inputs affect SR by altering both the fine root and microbial biomass in the community.

The similar response patterns of root respiration and microbial respiration were reasonable considering the close interactions between plants and microbes in biogeochemical cycles – the change in one could easily be transmitted to the other. However, the magnitude and pace of root respiration and microbial respiration responses differed. The N-induced reduction was greater for root respiration than for microbial respiration (see Fig. 4d, e). A likely reason is that, N decreased plant fine root biomass and the effect cascaded to the rhizosphere microbial community. Combined changes in root and rhizosphere microbial activities contributed to the observed high reduction in root respiration (because both root and rhizosphere microbial respiration changes were reflected in the measured root respiration using the trenching approach). Compared to root respiration, the N-induced increase in microbial respiration peaked at a lower N input level, and the abrupt transition of the decreased microbial respiration also occurred at a lower N input (Figs. 3, 4). Such differences may result from the overall higher N tolerance (and hence slower response to N) of plants compared to microbes[28]. The asynchronous responses of root respiration and microbial respiration were consistent with field observations, that in some experiments, N input decreased microbial respiration while still increasing root respiration[56].

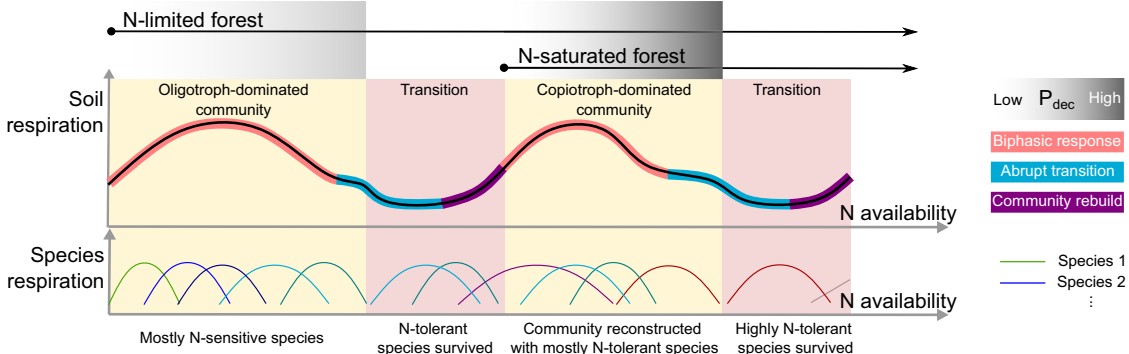

**Fig. 8 | A hypothetical framework for the responses of forest soil respiration to N inputs.** Initially, N input increases soil respiration in both N-limited and N-saturated forests, but the positive effect gradually diminishes with increasing N input (Figs. 3, 5), probably due to N toxicity and substrate availability constraints. In N-limited forests, a collapse of SR follows, probably caused by the transition from an oligotroph-dominated to a copiotroph-dominated community (Supplementary Text S2). Under increasing N availability, N-saturated forests can also experience a sudden collapse of soil respiration (Fig. 4). The abrupt transition is more likely to occur in N-saturated forests than in N-limited forests (Fig. 6), because N-saturated forests are initially under high N stress. At the bottom of the figure, the colored curves show the hypothesized responses of respiration at the species level. Species responses may add up to be the observed SR response to N inputs. The difference in community composition may explain the different SR responses in N-limited and N-saturated forests.

Plant and microbial properties may be important in explaining the spatial gradient of SR responses to N inputs. From boreal to tropical forests, the positive SR response tends to persist at increasingly high N inputs (Fig. 3j, k, l), indicating that tropical forests can respond positively to a wider range of N inputs (than boreal forests) before entering a collapse phase. Compared to N-limited forests, N-saturated forests respond positively to a relatively narrow range of N inputs (see Supplementary Fig. S3). This suggests that the background N availability of the forests may not be the main cause of differences in the SR response across biomes; otherwise, tropical forests with widespread N saturation would have shown a narrow response range to increasing N inputs. Instead, the differences may be attributable to the biodiversity gradient across the biomes[57,58], so species-rich tropical forests have a wider response range than boreal forests (Figs. 3, 8). N inputs could influence plant and microbial biomass and richness, thereby altering SR (Fig. 1). Despite the recognized importance of plant biomass and richness in N addition experiments[26], studies on microbial properties are relatively scarce. This contributes to the uncertainty surrounding how changes in microbial community structure and function would affect the SR response to N inputs. Future research aimed at interpreting the SR response factors (i.e., $f_{pos}$ and $f_{neg}$) and predicting the SR response in each forest community should consider investigating plant and microbial N status[59], as well as species richness, biomass, and N tolerance levels of plants and microbes.

Our analysis of plant root and microbial respirations was based on data collected using the root trenching method[60]. Because the root respiration data were not measured directly but inferred as the difference between soil respiration measured in the untrenched plot and that measured in the trenched plot, the potential error could be high. Once more root respiration data measured using other approaches (such as the isotopic method) become available, these data could be used to validate the patterns revealed in this study.

**N deposition enhanced forest soil respiration on a global scale**
We estimated the gross SR budget of global forests to be 34.4 PgC yr⁻¹ (Table 1), which should be interpreted as the average SR budget over the past half century during which SR fluxes have been observed in global forests. This budget is within previous estimates of 28.7–46.2 PgC yr⁻¹ [4,61,62,] (Supplementary Fig. S7). Based on the quantitative models and key parameters (i.e., $f_{pos}$, $f_{neg}$, $P_{dec}$) derived from global N addition experiments, we calculated that current levels of N deposition overall increased the global forest SR budget by ~5% (Table 1). The net effect of N deposition was more pronounced in temperate forests (than in tropical or boreal forests), where 8.6% of the annual SR could be attributed to N deposition.

The hotspots of N deposition-induced changes in SR were predominantly located in the temperate forests in Eastern China, Western Europe, and the Eastern United States (Fig. 7). The overall response of SR to natural N deposition, however, seems to be weak in temperate forests (Fig. 5). Unlike tropical forests which are naturally N-saturated or boreal forests which are mostly N-limited[14], the N status of temperate forests are rather mixed[13] (Supplementary Fig. S8). Thus, the same level of N deposition could increase SR in some temperate forests whereas decreasing SR in others. The counterbalancing of these two effects may have led to the seemingly weak SR response in the temperate biome (Figs. 3–5), although the spatial variation of SR responses is high there (Fig. 7).

In addition to the complex spatial pattern of forest N saturation status (which was unknown until recently)[13,14] and the lack of a general framework to explain the SR responses, the previous controversy about the effects of N deposition on global forest SR may be partly attributable to the design of N addition experiments and the use of the derived data.

Many N addition experiments have been preferentially conducted in temperate forests close to human settlements. Such forests, however, can be impacted by human activities and become N-saturated[13], leading to a biased SR response in the meta-analysis if differences in background N availability are not taken into account. Future N addition experiments should consider establishing new sites in intact forests in regions such as the central Amazon rainforest, the Congo rainforest and high-latitude taiga forests, thereby increasing data from these underrepresented regions. Also, the N addition rates used in experimental forests (typically no lower than 50 kgN ha⁻¹ yr⁻¹) are much higher than real-world atmospheric deposition rates (mostly below 20 kgN ha⁻¹ yr⁻¹ in global forests). Low N additions (in the range of 0–20 kgN ha⁻¹ yr⁻¹) should be adopted in experiments that aim to reveal the true response of SR to natural N deposition. In order to test the nonlinear SR response framework proposed here, more than three N addition levels may be needed, even at the expense of some experimental replication[63]. Moreover, the plant and microbial properties may be altered by long-term experimental N additions, resulting in forest SR deviating from its natural response to N deposition (Fig. 6). The short-term and long-term experimental data should be used to reveal the short-term response and long-term adaptation of SR to changes in N deposition, respectively.

This study revealed that the same level of N deposition can increase or decrease SR. The higher probability of SR being decreased by N deposition in N-saturated forests (Fig. 6) implies that the SR responses may be more diverse there than in N-limited forests. Experimental data confirmed the higher variability (and lower stability) of N-induced SR changes in N-saturated forests (Supplementary Fig. S9). That implies, high anthropogenic N deposition and N saturation in forests may have increased the variability of SR on a global scale, potentially affecting the stability of the global carbon cycle. In recent years, improved fertilizer management practices and reduced vehicle emissions have led to reductions in anthropogenic N deposition in the Eastern United States, Europe[9], and China[10,64]. Yet, these regions remain global hotspots for N emissions and deposition. If N pollution regulations could be maintained or further improved, N deposition levels and N-saturated forests would be expected to decrease, potentially reducing the forest soil respiration and its variability, thereby reducing the uncertainties in the projected carbon dynamics and future climate change.

## Methods
### Data sources
We systematically compiled annual soil respiration fluxes observed in forests worldwide where N addition experiments were conducted, by searching for relevant literature published before January 1, 2022 in the Web of Science Core Collection (www.webofscience.com) and the China National Knowledge Infrastructure Theses and Dissertations Database (https://oversea.cnki.net/kns?dbcode=CDMD). These databases are often used in global meta-analysis[21,65,66]. The search terms used were: forest AND (greenhouse gas OR $CO_2$ OR carbon dioxide OR soil respiration) AND (nitrogen OR fertiliz*). Then, we manually refined the retrieved 9417 studies based on the following criteria: (i) N addition experiment was conducted in the forest ecosystem, with records of site location and N addition dose; (ii) soil $CO_2$ flux was observed in the field and measured using gas chromatography, infrared gas analyzer, or alkali absorption techniques; (iii) where available, we also compiled data on root respiration and microbial respiration measured using the root trenching method. The compiled $CO_2$_exp dataset contains 1112 observations from paired control and experimental plots at 168 sites (Fig. 2).

To estimate the global forest soil respiration budget, we used soil respiration data observed under natural conditions (without manipulative experiments) from the global Soil Respiration DataBase (SRDB)[41]. In line with the $CO_2$_exp dataset, we only used data that met the following criteria: (i) soil $CO_2$ fluxes were observed in forest ecosystems with records of site location; (ii) soil $CO_2$ fluxes were observed in the field and measured using gas chromatography, infrared gas analyzer, or alkali absorption techniques. The derived $CO_2$_obs dataset contains 3689 observations from 966 forest sites worldwide (Fig. 2).

In addition, we collected auxiliary information on environmental factors (climate, N deposition, soil properties) from the same literature. For sites where not all the required information was available, we extracted data from spatial datasets based on site coordinates. Specifically, temperature and precipitation data were obtained from the Climatic Research Unit, University of East Anglia (https://crudata.uea.ac.uk/cru/data/hrg/cru_ts_4.03/). Soil texture data were from the Harmonized World Soil Database (https://www.fao.org/soils-portal/data-hub/soil-maps-and-databases/harmonized-world-soil-database-v12/en/). Soil chemistry data were from a published dataset by Shangguan et al.[67]. Global N deposition data were from a published dataset by Ackerman et al.[9]. Forest cover data were from the GLASS-GLC project[68]. Forest biome information was extracted from a spatial dataset of the Global Forest Monitoring project[69].

### Response of increased soil respiration to N input
Due to the different mechanisms dominating the positive and negative responses of SR to N input, the increased and decreased SR by N input may follow the biphasic and abrupt transition response patterns, respectively (Fig. 1; Supplementary Fig. S1). Therefore, we separately analyzed observations where N input increased or decreased SR. In our analysis, to avoid over-representation of data from the same experiment, we summarized data from the same site-year with the same N addition level by taking their averages. Each unique site-year-N addition level combination was used as a data point when we calculated the response factors of SR to N addition. The potential non-independence of the experimental data points was addressed indirectly by using an alternative effective size metric (i.e., the integrated response factor) and by running the analysis on subgroups of the dataset (See Supplementary Text S3 for details).

For each observation of increased SR (i.e., higher SR in N addition plot than in the corresponding control plot), we calculated a positive response factor ($f_{pos}$) to represent the effect size of the increase in SR due to N input (see Supplementary Text S1 for explanation and illustration of the response factors). The relationship between $f_{pos}$ and N input rate was analyzed for forests in each biome (i.e., tropical, temperate, and boreal) and in each N status (i.e., N-limited, N-saturated) or long-term experimental forest.

Throughout the analyses, the N-limited forest dataset ($CO_2$_exp_NL) consisted of experimental data from initially N-limited forests where N addition experiments had been conducted for no more than 3 years; the N-saturated forest dataset ($CO_2$_exp_NS) consisted of experimental data from initially N-saturated forests where N addition experiments had been conducted for no more than 3 years; the long-term experimental forest dataset ($CO_2$_exp_LT) contained data from forests where N addition experiments had been conducted for more than 3 years, regardless of the initial N status. This is because long-term N addition could have resulted in accumulated and exceptionally high N availability, overriding the initial N status difference of the sites. Previous studies have also suggested different biotic responses in N addition experiments lasting for less than or more than 3 years[70–72]. The N limitation or saturation status of global forests was determined using the sensitivity of soil $N_2O$ emission to N deposition as an indicator[13]—N-saturated forests have higher sensitivity of soil $N_2O$ emission to N deposition than N-limited forests because of the relatively open N cycle.

**Positive response factor ($f_{pos}$).** The positive response factor of soil respiration to N input ($f_{pos}$; $kgCO_2$ $kgN^{-1}$) was designed to reveal the increased SR per unit of artificial N addition (Fig. 2e). The observed $f_{pos}$ was calculated using experimental data:

$$f_{pos,\,obs} = \frac{\Delta SR_{N-inc}}{\Delta N} = \frac{(SR_{+N} - SR_{CL})}{N_{add}} \qquad (1)$$

In the equation, $SR_{CL}$ ($kgCO_2$ $ha^{-1}$ $yr^{-1}$) was the soil respiration rate in the control plot without artificial N addition, and $SR_{+N}$ ($kgCO_2$ $ha^{-1}$ $yr^{-1}$) was the soil respiration rate observed at the same site under an artificial N addition rate of $N_{add}$ ($kgN$ $ha^{-1}$ $yr^{-1}$). $\Delta SR_{N-inc}$ was the N-induced increase in soil respiration.

**Increased soil respiration by N input.** It is difficult to directly quantify the biphasic (quadratic) relationship between the increased SR and N input, especially when the change point(s) in the relationship are uncertain. In addition, the differences between sites could further complicate the analysis (Supplementary Fig. S10). Therefore, we used a derivative method and analyzed the potentially linear relationship between the positive response factor ($f_{pos}$; which is like the sensitivity of increased SR to N input) and N input rate. Specifically, we built linear models to predict $f_{pos}$ of forests (one for each biome and each N

status):

$$f_{pos,\,pred} = c_1 \times N_{input} + c_2 \qquad (2)$$

Here, $c_1$ and $c_2$ were characteristic parameters of the linear models estimated using the lm function in R[73]. $N_{input}$ represented any N input rate (unit: kgN ha$^{-1}$ yr$^{-1}$), which was the sum of atmospheric N deposition ($N_{dep}$) and artificial N addition ($N_{add}$).

Because $f_{pos}$ was similar to the sensitivity of increased SR to N input, it was then inferred from Eq. 1 and Eq. 2 that the increased SR by N input could be quantified as

$$\Delta SR_{N-inc} = SR_{+N} - SR_{CL} = f_{pos,\,pred} \times N_{add} = \left( c_1 \times N_{input} + c_2 \right)$$
$$\times \left( N_{input} - N_{dep} \right) = c_1 \times \left( N_{input} \right)^2 + \left( c_2 - c_1 \times N_{dep} \right) \times N_{input} - c_2 \times N_{dep} \qquad (3)$$

For each site, $c_1$, $c_2$, and $N_{dep}$ were nearly constant, and $N_{input}$ was the variable in Eq. 3. Therefore, the linear relationship between $f_{pos}$ and N input rate could be translated to the quadratic (and inverted U-shaped when $c_1$ was negative) relationship between the increased SR and N input rate. The inverted U-shaped curve is a typical response curve indicating a biphasic effect.

If there was no N deposition or artificial addition (i.e., N input rate was 0), $f_{pos,\,pred} = c_2$. It was inferred that the increased soil respiration by N deposition in a forest could be quantified as

$$\Delta SR_{N-inc} = SR_{CL} - SR_0 = - \left( f_{pos,\,pred} \times (0 - N_{dep}) \right) = c_2 \times N_{dep} \qquad (4)$$

where $SR_0$ (kgCO$_2$ ha$^{-1}$ yr$^{-1}$) was the "background" soil respiration rate when there was no external N input (natural N deposition or artificial N addition), $SR_{CL}$ was the soil respiration rate observed in the control plot receiving natural N deposition (kgCO$_2$ ha$^{-1}$ yr$^{-1}$).

### Response of decreased soil respiration to N input

Decreased soil respiration by N input may result from an abrupt transition in the community structure and function, the magnitude of which may be independent of the N input rate that triggers the transition. Therefore, we quantified a negative response factor ($f_{neg}$; see Supplementary Text S1 for explanation and illustration of the response factors) to represent the magnitude of the reduction.

**Negative response factor ($f_{neg}$).** The negative response factor of soil respiration to N input ($f_{neg}$) was calculated for each N addition plot that observed a decreased SR as compared to the control plot. $f_{neg}$ represents the effect size of the decrease in SR due to N input. Observed $f_{neg}$ was calculated for forests in each biome and each N status using experimental data:

$$f_{neg,\,obs} = \frac{\Delta SR_{N-dec}}{SR_{CL}} = \frac{(SR_{+N} - SR_{CL})}{SR_{CL}} \qquad (5)$$

where $SR_{+N}$ was the soil respiration rate observed under each level of N addition (unit: kgCO$_2$ ha$^{-1}$ yr$^{-1}$), $SR_{CL}$ was the soil respiration rate observed in the control plot (unit: kgCO$_2$ ha$^{-1}$ yr$^{-1}$), $\Delta SR_{N-dec}$ was the N-induced decrease in soil respiration (unit: kgCO$_2$ ha$^{-1}$ yr$^{-1}$).

**Decreased soil respiration by N input.** If $f_{neg}$ does not change gradually with N input, but instead changes abruptly at a certain point, then it can be inferred that the decrease in soil respiration is not a dose-dependent, gradual process, but rather an abrupt transition triggered by a certain level of N input.

If $f_{neg}$ remains relatively constant within a certain level of low N input, the decreased soil respiration by N deposition in a forest could

be quantified as

$$\Delta SR_{N-dec} = SR \times f_{neg,\,obs} \qquad (6)$$

where SR was the soil respiration rate under natural N deposition and climatic conditions (the estimation of the SR of global forests is detailed in a later section), $f_{neg}$ was the negative response factor corresponding to low N inputs (Eq. 5).

### Integrated response of soil respiration to N input

To support our separate analysis on the increased and decreased SR by N inputs, we also did an integrated analysis without distinguishing the positive and negative responses of SR. The algorithm for calculating the integrated response factor ($f_{int}$) has been used in our previous research[15]. This method, however, can only qualitatively reveal the trend of how SR changes in response to increasing N inputs.

**Integrated response factor ($f_{int}$).** The integrated response factor ($f_{int}$; kgCO$_2$ kgN$^{-1}$) was calculated to reveal the change in SR (positive or negative) within a given range of N input. $f_{int}$ was calculated as

$$f_{int} = \frac{(SR_2 - SR_1)}{(N_2 - N_1)} \qquad (7)$$

where $SR_1$ and $SR_2$ (unit: kgCO$_2$ ha$^{-1}$ yr$^{-1}$) were the soil respiration rates observed under two different N input rates of $N_1$ and $N_2$ (kgN ha$^{-1}$ yr$^{-1}$), respectively. To make the $f_{int}$ comparable across forest sites, $f_{int}$ was calculated within four defined N input ranges, $\leq$ 50 kgN ha$^{-1}$ yr$^{-1}$, 50–100 kgN ha$^{-1}$ yr$^{-1}$, 100–150 kgN ha$^{-1}$ yr$^{-1}$, and $\geq$ 150 kgN ha$^{-1}$ yr$^{-1}$.

We calculated the integrated response factors using SR observations from experimental N addition forests that were initially N-limited (CO$_2$_exp_NL dataset in the main text). Initially N-saturated forests were not used in this analysis because only the N-limited forests would go through the entire process (including the community collapse and reconstruction phase) and show a bimodal response pattern under increasing N inputs. We then compared the integrated response patterns of SR to N inputs with the patterns revealed in our previous separate analysis of positive and negative responses (see Supplementary Text S2 for details).

### Random forest regression model for estimating global forest soil respiration rate

Using soil respiration (SR) data observed under natural conditions (CO$_2$_obs dataset), we predicted the SR of each forest-covered grid (0.5° × 0.5°) using the random forest regression method[74]. In practice, multiple SR observations from the same site-year were aggregated by taking the mean value. After excluding outliers that were at least three interquartile ranges away from the global median, we randomly sampled 20% of the data to form a test dataset. 75% of the remaining data (i.e., 60% of the total data) were randomly selected for training the random forest models, and the remaining data were used to allow the variation in the training dataset (to reduce the influence of random sampling on the derived models). Climate, N deposition, soil texture, soil organic carbon, soil nitrogen content, soil pH, and N status variables were used as predictors (Supplementary Table S4). These variables were selected based on data availability and mechanistic relevance[50,75–79].

Because the constructed models can vary depending on the data used to train the models, the random sampling of the training data was repeated 1000 times, deriving 1000 models. When estimating SR at the grid level, each grid had 1000 predicted SR values from the 1000 models. The mean SR of the 1000 values was used as the estimated SR of the grid. Estimated SR for grids in the test dataset, which were never

used in model construction, were then compared with observed values to measure the accuracy of prediction.

Meanwhile, we estimated the partial dependence of SR on N deposition using each of the 1000 random forest models. The derived partial dependence curves depict the change in SR across the natural gradient of N deposition, which could potentially validate the responses of SR to N input that we estimated using experimental data. In the partial dependence analysis, N deposition-induced SR change ($\Delta SR_N$) was estimated as the difference in SR between forests with N deposition rates below 1 kgN ha$^{-1}$ yr$^{-1}$ and forests with N deposition rates above the level, thus removing the influence of basal SR differences across biomes and N status.

Similarly, we built random forest models for the negative response factor ($f_{neg}$) using the experimental dataset ($CO_2$_exp) with 1000 repetitions. This analysis was not for prediction, but to assess the relative importance of environmental factors on $f_{neg}$, and to estimate the partial dependence of $f_{neg}$ on N input rate.

We used the randomForest package[80] in R[73] for the random forest regression analysis, the relative importance analysis, and the partial dependence analysis. Random forest model parameters were optimized using tuneRF function in the same package. The segmented package[81] was used to detect the change points in the partial dependence curves.

## Probability of soil respiration being decreased by N input ($P_{dec}$)

The proportions of forests where experimental N inputs decreased SR were calculated to model the probability of SR being decreased ($P_{dec}$) by N inputs. $P_{dec}$ as a regular, predictable parameter was then applied to quantify the proportion of forests where SR was decreased by natural N deposition.

The same N input level may decrease SR in one forest while increasing SR in another. Based on the $CO_2$_exp_NL dataset, we summarized the frequencies of each N input rate decreasing SR in N-limited forests. A nonlinear regression model was then constructed to predict the probability of soil respiration being decreased by N inputs ($P_{dec}$) in N-limited forests. The same processes were carried out using the $CO_2$_exp_NS and $CO_2$_exp_LT datasets, deriving models for predicting $P_{dec}$ in N-saturated and long-term experimental forests, respectively. On a regional scale, because there are plenty of forests, the proportion of the forests where a N input level decreases SR approximates $P_{dec}$ (Law of Large Numbers).

The frequency of SR being decreased by a given N input rate in N addition experiments was the observed $P_{dec}$. That is,

$$P_{dec,obs} = \frac{n(\Delta SR_N < 0)}{n(\Delta SR_N)} = \frac{n(SR_{+N} - SR_{CL} < 0)}{n(SR_{+N} - SR_{CL})} \quad (8)$$

In the equation, $n(\Delta SR_N < 0)$ was the number of observations that SR was decreased by a given level of N input. $n(\Delta SR_N)$ was the total number of N-induced SR changes ($\Delta SR_N$) corresponding to the N input level. However, the $\Delta SR_N$ values corresponding to a given N input rate may be few in number. To reduce the potential bias due to insufficient observations, we counted all $\Delta SR_N$ values corresponding to N input rates that were no more than 10 kgN ha$^{-1}$ yr$^{-1}$ different from the analyzed N input level. For example, $P_{dec,obs}$ corresponding to 20 kgN ha$^{-1}$ yr$^{-1}$ was calculated using $\Delta SR_N$ corresponding to the N input rates ranging from 10 to 30 kgN ha$^{-1}$ yr$^{-1}$.

By fitting models to $P_{dec,obs}$, we could predict $P_{dec}$ for any N input rates. $P_{dec}$ as a probability should be between 0 and 1, so we only tried to fit regression models with lower and upper bounds. After comparing the performance of several different types of regression models (including log-logistic model, Weibull model, Michaelis–Menten model) based on Akaike information criterion (AIC; Supplementary Table S6), we used the Michaelis–Menten-type model, shown in Eq. 9, to predict $P_{dec}$.

$$P_{dec,pred} = \frac{N_{input}}{c_3 + N_{input}} \quad (9)$$

where $c_3$ was the characteristic parameter of this nonlinear regression model, estimated using the drc package in R[82]; $N_{input}$ represented any N input rate (unit: kgN ha$^{-1}$ yr$^{-1}$).

## Quantifying the contribution of N deposition to global forest soil respiration

Using Eq. 4 and Eq. 6, we could estimate the increased or decreased SR by N deposition under any N deposition level. Using Eq. 9, we could estimate the proportion of forests where SR was decreased by a N deposition rate in a grid (0.5°× 0.5°, which contains many forests). On the basis, we quantified the effect of N deposition on forest SR at the grid level (Eqs. 10–12). Summarizing the grid-level data, we obtained the annual SR in global forests, the N deposition-induced change of SR, and the contribution of N deposition to the global forest SR budget (Eq. 13):

$$\Delta SR_{N-inc,i} = c_{2,i} \times N_{dep,i} \times (1 - P_{dec,i}) = c_{2,i} \times N_{dep,i} \times \left(1 - \frac{N_{dep,i}}{c_{3,i} + N_{dep,i}}\right) \quad (10)$$

$$\Delta SR_{N-dec,i} = SR_i \times f_{neg,i} \times P_{dec,i} = SR_i \times f_{neg,i} \times \frac{N_{dep,i}}{c_{3,i} + N_{dep,i}} \quad (11)$$

$$\Delta SR_{N,i} = \Delta SR_{N-inc,i} + \Delta SR_{N-dec,i} \quad (12)$$

$$\text{Contribution rate} = \frac{N - \text{induced change}}{\text{Budget}} = \frac{\sum_i(\Delta SR_{N,i} \times Area_i)}{\sum_i(SR_i \times Area_i)} \times 100\% \quad (13)$$

where $\Delta SR_{N,i}$ represented the N deposition-induced change in forest soil respiration in grid i (kgCO$_2$ ha$^{-1}$ yr$^{-1}$); $\Delta SR_{N-inc,i}$ represented the increased soil respiration by N deposition in grid i (kgCO$_2$ ha$^{-1}$ yr$^{-1}$); $\Delta SR_{N-dec,i}$ represented the decreased soil respiration by N deposition in grid i (kgCO$_2$ ha$^{-1}$ yr$^{-1}$), SR$_i$ was the soil respiration rate in grid i (kgCO$_2$ ha$^{-1}$ yr$^{-1}$), Area$_i$ was the forest area in grid i (ha).

Data analysis in this study was performed using R[73]. The level of significance was set at $p < 0.05$ unless otherwise noted. Maps were generated using ArcGIS software[83].

## Data availability
Data supporting the findings of this study (including $CO_2$_exp and $CO_2$_obs datasets) are available in Zenodo (https://doi.org/10.5281/zenodo.17670031).

## Code availability
R code file supporting the findings of this study is available in Zenodo (https://doi.org/10.5281/zenodo.17670031).

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

## Acknowledgements

The authors thank Prof. James Raich (Iowa State University) for his inputs on early version of this paper. We are grateful to all the researchers who spent time and effort on manipulative experiments and observations and made the data publicly accessible. This work was financially supported by the National Natural Science Foundation of China (32430067, 32588202, 42141004) and the National Key R&D Program of China (2023YFF1305900, 2022YFF080210102) received by N.H., and the Pioneer Center for Landscape Research in Sustainable Agricultural Futures (Land-CRAFT), DNRF grant number P2 received by K.B.B.

## Author contributions

X.C. collected data and carried out the analysis based on feedback from N.H., P.V., and K.B.B. X.C. drafted the initial manuscript, P.V., N.H., B.B.L., S.N., E.D., K.Y., M.Z., K.V.S., E.L.P., L.H., L.X., M.L., and K.B.B. reviewed and edited the manuscript.

## Competing interests

The authors declare no competing interests.
