## [Transparent Peer Review file · Nature Communications]

A general framework for nitrogen deposition effects on soil respiration in global forests

Corresponding Author: Professor Nianpeng He

Version 0:

Reviewer comments:

Reviewer #1

(Remarks to the Author)

This study examines the effects of N deposition on forest soil respiration using both data from N addition experiments and observations under real atmospheric N deposition. The approach of integrating these two datasets to compare soil respiration responses is certainly a valuable idea. However, at this stage, I have several major and fundamental concerns regarding the hypotheses and statistical analyses that need to be addressed before I can fully assess the details of the manuscript or recommend it for publication.

1. The unclear Conceptual Framework: The authors dedicate considerable discussion and three figures to presenting a hypothesized framework for forest soil respiration responses to N inputs. While this conceptual framework aimed at helping understanding the SR response patterns and mechanisms, I did not find sufficient theoretical support or empirical evidence in the introduction or relevant literature to substantiate this framework. For instance, in Lines 330–332, the authors mention a "transition from an oligotroph-dominated to a copiotroph-dominated community," but this concept is not clearly introduced or explained in the introduction. Furthermore, the study's results do not appear to provide strong support for this framework—most of the quadratic relationships shown in Figure 3 are not statistically significant, making it difficult to justify a biphasic response pattern. Overall, I find it difficult to be convinced by this new framework. Besides, even though the framework is illustrated in three figures, this level of redundancy does not compensate for the lack of mechanistic support. Additionally, Figure 1 does not effectively aid in understanding why a biphasic and abrupt response would be expected.
2. It is unclear why the authors chose to analyze positive and negative SR responses separately. SR response to N deposition is a continuous variable that can range from positive to negative within the same experiment. While I agree that different pathways can drive either positive or negative responses, this does not necessarily mean that these responses should be analyzed in isolation. It is more likely that both positive and negative mechanisms operate simultaneously, with the net response depending on which mechanisms dominate under a given N deposition rate and background conditions. Without a clear theoretical justification for this separation, this analytical approach seems somewhat arbitrary.
3. Lines 526–535: The distinction between "N-limited," "N-saturated," and "long-term" experimental forests is not clearly justified. In particular, the rationale for defining three years as the threshold for high N stress is unclear—why not two, five, or ten years? Moreover, N stress is not solely determined by duration; the N addition rate also plays a critical role. For example, a low N addition rate over a longer period (e.g., 10 kg ha⁻¹ yr⁻¹ for four years) may impose less stress than a high N addition rate over a shorter period (e.g., 100 kg ha⁻¹ yr⁻¹ for three years), given differences in total N input. This issue requires further clarification and justification.
4. For the data derived from N addition experiments, it is essential to clarify how potential non-independence among data points within each original study was accounted for in the statistical analyses. Addressing this concern would help ensure the robustness of the findings.
5. Use of Abbreviations: The manuscript introduces and relies on numerous abbreviations, many of which are not clearly defined, making the text difficult to follow. In several cases, I only fully understood their meaning after referring to the Methods section. I recommend using abbreviations only when necessary and ensuring that all abbreviations are explicitly defined in the introduction to improve readability.

(Remarks on code availability)

Reviewer #2

(Remarks to the Author)

Reviewer Comments to the Author:

Cen et al. systematically compiled and analyzed a dataset of annual soil respiration fluxes observed in forests worldwide where nitrogen addition experiments were conducted. Meanwhile, they applied a natural gradient approach to analyze the SRDB database and the global nitrogen deposition dataset. Based on these analyses, they proposed a general framework for the effects of nitrogen deposition on soil respiration in global forests. Subsequently, they assessed the net effect of nitrogen deposition on the global forest soil respiration budget. The proposed general framework for the response of soil respiration to nitrogen inputs is innovative. In addition, they have also made several new estimations of large-scale indicators. I have a few suggestions for your consideration.

First, it is important to note that N addition typically promotes soil respiration under conditions where nitrogen saturation has not yet been reached. If a forest is already in a nitrogen-saturated state, the more plausible outcome would be nitrogen addition primarily leading to inhibition of soil respiration. In most ecosystems, nitrogen addition initially stimulates soil respiration, followed by a suppression phase, and eventually leads to a collapse, resulting in a new equilibrium. Once this equilibrium is reached, the ecosystem becomes nitrogen-unsaturated, and this cycle may repeat. I recommend that the authors clarify this point in the context of nitrogen saturation and its role in the nitrogen addition experiments discussed in the manuscript.

Second, in multi-level N addition single-point experiments, if the data can be fitted to a unimodal curve, it would be helpful to assess whether the majority of experimental fits exhibit a unimodal pattern. Such evidence could help reduce the noise caused by heterogeneity between experiments, thereby strengthening the reliability of the conclusions drawn in the manuscript. Although the dataset is relatively small, I recommend that the authors attempt to provide more substantial evidence to support their findings, which would enhance the robustness of the study and its conclusions.

Finally, this study involves numerous estimations and predictions, making the methods for model construction and the reliability of predictions crucial. I recommend that the authors provide more detailed and reliable explanations, as outlined in the Specific Comments section below.

Specific Comments:

Line 53-55: "High exogenous N inputs (deposition or fertilization) "and" long-term N additions" are somewhat repetitive." You should clarify what you want to express.

Line 56-57: In my opinion, nitrogen availability refers only to the amount of nitrogen in an ecosystem that is available for biological use.

Line 60-61: "background N stress" It should be the availability of nitrogen, not every forest is under N stress.

Line 66: "Soil respiration (SR)" Rs is more commonly used as the abbreviation for soil respiration.

Line 69: "mixed effects" diverse effects

Line 112-114: The studies by Janssens, I. A. et al. and Zhou, L. Y. et al. are both based on meta-analysis results, revealing the overall effects of numerous studies, many of which show that nitrogen addition promotes soil respiration. However, the entry point you mentioned, that low nitrogen promotes soil respiration, seems inconsistent with their findings, and this point comes across as somewhat confusing.

Line 134-136: The current phrasing may give the impression that the biphasic dose-response relationship is solely due to soil acidification. It is recommended to separate the explanation and the summary for better clarity.

Line 187-189: Rs, Ra, and Rh should be the commonly used abbreviations for soil respiration and its components.

In addition, I have a question: The effect size of such an abrupt transition is often disproportionate to the N addition level triggering the change. How does the algorithm used in this paper differ from the response ratio commonly used in meta-analysis? This is just a curiosity and not a challenge to your calculation method.

Line 240, 248: The change in soil respiration (Δ SRN) should be explained in more detail, as I am unclear about how the change is calculated up to this point.

Line 265-266: The classification of "N-limited forests," "N-saturated forests," and "long-term experimental forests" is confusing, as it is unclear why "long-term experimental forests" should be considered separately.

Line 326-327: I have been unable to understand the meaning of Pinh in Fig. 7. According to this framework, the results from the N-limited and N-saturated forest datasets are the most important evidence and should be highlighted in the main text.

Line 353: In N-limited forests, the old communities transition into new communities. Does this mean that the new communities represent N-saturated forests? Will their nitrogen demand increase?

In Appendix Text S2, with increasing N input, the local response factor (fb) of SR to N input shifted from positive to negative and then back to positive in N-limited forests across all biomes (Fig. S5).

Line 390: The two explanations for nitrogen addition inhibiting soil respiration should not be seen as an incremental relationship like "even".

Line 417-419: The theme of this paragraph is to discuss the similarities and differences across different ecosystems and nitrogen status categories. The mention of response range here seems somewhat out of place.

Line 422: The explanation, i.e., from N-limited to N-saturated to long-term experimental forests, might be more helpful to the reader if mentioned earlier in the text.

Line 451-455: This content should be placed at the end in the conclusion and significance section. It might serve as a transition to the next section, but it feels somewhat abrupt when read.

Line 451-482: It is recommended to organize and categorize these insights and limitations. As it stands, the reading feels somewhat disorganized and chaotic.

Lines 499-505: Did you use only the SRDB data, or did you also include the CK data from your own CO₂_exp dataset?

Lines 608-609: What was the basis for selecting the predictor variables? Why were these specific variables ultimately chosen?

Lines 600: The selection and determination of random forest model parameters (e.g., Ntree, Mtry, and nodesize) have a significant impact on model performance. The authors should clarify how this issue was addressed in the Methods section. The soil respiration data were likely collected over different time periods, leading to potential inconsistencies. How did the authors account for the influence of study duration in their analysis?

The study by Ni Huang, Spatial and temporal variations in global soil respiration and their relationships with climate and land cover, has also conducted global-scale soil respiration predictions, achieving an R² of 0.6–0.8. In contrast, the R² in this study is only 0.25. How reliable are the predictions? Have the authors considered alternative machine learning approaches, such as support vector machines (SVM), to improve model performance?

Lines 651-658: What was the basis for selecting this nonlinear regression model? Is it the optimal model? The authors should provide a more detailed explanation, as the current description remains unclear.

Using this empirical equation for prediction does not seem to be theoretically superior to machine learning. Why not use machine learning to predict Pdec for any N input rates?

(Remarks on code availability)

The code execution is reliable.

Reviewer #3

(Remarks to the Author)

Overall, this is a timely manuscript in which the authors sought to determine the net effect of atmospheric nitrogen deposition on rates of soil respiration in forests around the globe. They conducted a meta-analysis with data compiled through published papers cited in Web of Science and from the China National Knowledge Infrastructure Theses and Dissertations Database.

Here are my main constructive critiques of the paper:

Their dataset is biased toward Chinese collections given they looked at papers cited in Web of Science, but only dissertations from the “China National Knowledge Infrastructure Theses and Dissertations database”. It’s unclear why they used student papers from China and not elsewhere.

There are many places where the writing needs to be improved. For example, in the Highlights it says “Experimental and observational data agree on a new framework...” As written, it sounds like the data themselves are thinking and agreeing on something; the authors need to make it clearer that researchers can agree on a new framework, data cannot. This and other text throughout the paper should be revised in this way.

In general I suggest using fewer abbreviations. Many are not necessary. For example, why is “Pdec” used throughout and not just “N inputs”? If the authors want this paper to be read and understood by a broad audience, it would be much easier to read and follow if fewer abbreviations are used.

There are several places where the authors make a comparison, but don’t finish it. See below for details.

The following detailed comments are meant to improve the manuscript:

Highlights

Line 42: Change first bullet to be less anthropomorphic

Line 44: Change to “N deposition reduces...” and finish the statement “more frequently in N-saturated than non-N saturated”? Or “more frequently than “N-limited forests”? Or something else?

Line 46: “Responses of plant root and microbial respirations to N input differ in magnitude and pace”

- I suggest replacing this with the actual differences.

Line 47: “Current level of N deposition enhanced global forest soil respiration by ~5%” relative to what? Pre-industrial levels of atmospheric deposition? Non-forest ecosystems?

Abstract

Lines 73-76: This text is unclear “Using a novel probabilistic approach, we estimated that N deposition decreased SR in 2.9% of global forests, mostly N-saturated forests in eastern China, western Europe, and eastern USA. But the net effect of N

deposition increased the global forest SR budget by 5.1% (1.7 PgC yr⁻¹)." The title of the paper includes these three regions only (eastern China, western China, and eastern USA); what regions are the authors referring to about the "net effect of N deposition"?

Lines 73-75: Also related to this text: "Using a novel probabilistic approach, we estimated that N deposition decreased SR in 2.9% of global forests, mostly N-saturated forests in eastern China, western Europe, and eastern USA."

-Are the authors saying that N deposition increased SR in the non N-saturated forests of eastern China, Western Europe, and eastern USA?

-Or, are they saying that N deposition increased SR in other regions of the world?

Here and elsewhere the authors often state the first half of a comparison, but don't complete it so it's difficult to assess the assertion they are making

Lines 76-77: "If N pollution could be effectively controlled, global forest SR and its variability would decrease, thereby reducing the uncertainty in the projected terrestrial carbon dynamics."

-It's unclear what evidence the authors are using to draw this final conclusion.

-I suggest they share results about variability and uncertainty before jumping from trends in N deposition effects on SR (Lines 73-76) to how reductions in N deposition would impact variability in SR and our predictions of this important process (Lines 76-77)

Line 70: Include the total number of experiments included in the data synthesis

Introduction

Line 80: Spell out "carbon dioxide" first time CO₂ is used

Lines 148-149: The authors need to make clear what the experimental unit is here. Why 1112 observations, but only 168 experiments? Did they keep data separate by tree species? Form of N addition? How did they handle time series data? Take averages over time before including in the data synthesis?

Lines 156-159: Move this text to start the paragraph so the main goals are stated first, then followed by the methods (including # of observations and experiments included in the study)

Figure 3:

How were biomes defined? In other words, did the authors utilize type of biome noted by authors of published data? Did they compile latitude and longitude and categorize biomes themselves? Did they utilize mean annual precipitation and temperature?

It appears from Figure 3A-C that rates of total soil, root, and microbial respiration all decline with greater rates of experimental N inputs AND that the steepest declines occur at the lowest rates of N inputs (which is contrary to their hypotheses and conclusions). It is unclear why the authors describe these relationships as "increased SR by N input"

Figures 3A-C and G-I: I suggest showing the statistical results for both slopes (the red and gray lines in each)

I understand the detailed Methods come below, but the authors need to explain in the legend for Figure 3 what the experimental unit is (each dot). Is it site? Tree species kept separate? How are time series from individual sites handled?

Line 241: The authors state "Under high N deposition, ΔSRN decreased. The declining trends were more pronounced in tropical forests and N-saturated forests,..."

-However, it's unclear how they are designating sites as "N-saturated" or not. It should be briefly defined here

Lines 263-265: The authors state "Generally, P_{dec} increased with increasing N stress, which can result from high N inputs, forest N saturation, or long-term artificial N additions."

-It's unclear how the authors have defined "N stress" based on their results. It is clear conceptually that high N input sites that are N saturated can experience "stress", but it's unclear how the authors are utilizing their data to come to this conclusion.

-Is "stress" here defined as the "probably of forest soil respiration being decreased by N input"?

Line 303-304: The authors state "Globally, we found that current levels of N deposition increased SR in most forests, by a total of 1.8±0.1 PgC yr⁻¹."

-It would be stronger for the authors to report this in terms of kg N added through atmospheric inputs.

Lines 303-304 and Table 1: The results would be more intuitive and powerful if the authors expressed the reduction or increase in soil respiration caused by N inputs as something like: "PgC kg N inputs⁻¹ yr⁻¹"

Discussion (no line numbers provided by authors)

Methods Section (no line numbers provided by authors)

Section 4.1: The authors need to explain how they handled multiple samplings from the same study (1112 observations). Did they take the average across time so only one soil respiration rate for control and N-addition plots were included? Did they keep growing season from winter measurements separate or did they combine them?

(Remarks on code availability)

Bibw

Version 1:

Reviewer comments:

Reviewer #1

(Remarks to the Author)

I appreciate the authors' detailed responses and their efforts to clarify the rationale behind their analytical approach. However, I remain unconvinced by the decision to separate the analysis of positive and negative effects on SR, as no compelling theoretical or empirical justification has been provided to support this separation. Since this constitutes a fundamental step in their analytical framework, it casts doubt on the validity of the subsequent results and conclusions. In their response, the authors suggest that increased SR results from "N-facilitated biomass growth," while decreased SR arises from "N-induced biodiversity loss." However, numerous grassland experiments have demonstrated that both biomass increase and biodiversity loss can occur simultaneously under the same nitrogen treatments (e.g., Bai et al., 2009, *Global Change Biology*, <https://doi.org/10.1111/j.1365-2486.2009.01950.x>). This evidence supports the point I raised earlier: both positive and negative mechanisms may operate concurrently within the same plot, making it problematic to analyze them in isolation.

Also, the authors mentioned that the positive N effect on SR is biphasic because of "high N becomes toxic". I would argue that "high N becomes toxic" is also one of the reasons for negative responses of SR to N addition. This again challenges the validity of separating these effects analytically.

Overall, I find the separation of positive and negative SR responses to nitrogen addition conceptually confusing and methodologically unsubstantiated. Without stronger evidence or theoretical support for this approach, I cannot endorse the results or conclusions that depend on it.

(Remarks on code availability)

Reviewer #2

(Remarks to the Author)

The authors have made thorough and appropriate revisions in response to the Editor's and reviewers' previous comments. I am satisfied with the current version of the manuscript and have no further comments.

(Remarks on code availability)

The code execution is reliable.

Response to reviewers' comments:

Reviewer #1 (Remarks to the Author):

#2. This study examines the effects of N deposition on forest soil respiration using both data from N addition experiments and observations under real atmospheric N deposition. The approach of integrating these two datasets to compare soil respiration responses is certainly a valuable idea. However, at this stage, I have several major and fundamental concerns regarding the hypotheses and statistical analyses that need to be addressed before I can fully assess the details of the manuscript or recommend it for publication.

1. The unclear Conceptual Framework: The authors dedicate considerable discussion and three figures to presenting a hypothesized framework for forest soil respiration responses to N inputs. While this conceptual framework aimed at helping understanding the SR response patterns and mechanisms, I did not find sufficient theoretical support or empirical evidence in the introduction or relevant literature to substantiate this framework. For instance, in Lines 330–332, the authors mention a "transition from an oligotroph-dominated to a copiotroph-dominated community," but this concept is not clearly introduced or explained in the introduction.

Response: We have revised the relevant text in the Introduction section (L105–116) and Fig. 1 to clearly introduce the N-induced change in community composition. Previous studies, such as Gilliam (2006) and Bobbink et al. (2010), reported that excessive N deposition could lead to a transition from a community dominated by species adapted to N-poor environments to one dominated by nitrophilic copiotrophs. We have included these references in the Introduction to better support the proposed conceptual framework.

Additionally, we clarified the theoretical basis of the proposed conceptual framework using text (L92–116 in the main text, Supp. Text S1) and figure (Supp. Fig. S1). There have been several studies to explain the effects of N deposition on forest soil respiration, such as Janssens et al. (2010) and de Vries et al. (2014). By proposing a new conceptual framework, we don't mean to undermine the previous theories, but to complement them – the inclusion of the community collapse-reconstruction mechanism into the traditional N saturation theory may help to better explain the N-induced increase in soil respiration in some N-rich forests and long-term experimental N addition forests, as we discussed in more details in the Discussion section of the revised manuscript.

#3. Furthermore, the study's results do not appear to provide strong support for this framework—most of the quadratic relationships shown in Figure 3 are not statistically significant, making it difficult to justify a biphasic response pattern. Overall, I find it difficult to be convinced by this new framework. Besides, even though the framework is illustrated in three figures, this level of redundancy does not compensate for the lack of mechanistic support. Additionally, Figure 1 does not effectively aid in understanding

why a biphasic and abrupt response would be expected.

Response: The quadratic relationship between N input and soil respiration has been mentioned or suggested in several previous studies, including Li et al. (2019), Wang et al. (2020), and Xing et al. (2022a). In this synthesis analysis, the lack of statistical significance of the quadratic relationships (in Fig. 3) may result from the different initial N availability and biotic properties in the experimental sites. Even though there is a quadratic relationship between N input and the increased soil respiration in each site, the differences in the peaks of the quadratic curves (which may depend on the nitrogen use efficiency of the biotic communities), the differences in the y-intercepts (which may depend on the initial N availability in the ecosystem), and the differences in the x-intercepts (which may depend on the biotic tolerance to high N availability before the community collapse) can lead to a relatively weak quadratic relationship when data from all sites were analyzed together. We added a Supp. Fig. S10 to graphically explain the potential causes that may weaken the quadratic relationships.

Fig. S10 Potential causes that may weaken the quadratic relationship between N inputs and increased soil respirations in a data synthesis analysis. In panels A–C, points of the same color show experimental data from the same site, and dashed curves of the corresponding color show the site-specific quadratic curve. The black solid curve shows the fitted quadratic curve using data from all sites. As can be seen, the goodness of fit of the black curves is not as good as that of the site-specific colored curves, due to the different peaks and intercepts of the site-specific curves. However, with increasing N input, the overall response factor of increased soil respiration to N input (f_a ; see Supp. Fig. S1 for detailed explanation and illustration) decreases regardless of whether the model was fitted for each site or for all sites combined (see the change from f_a to f_a' in the panels). Therefore, a significant negative relationship between f_a and N input was found in all forests (Fig. 3A–C, G–I in the main text), despite the differences between sites.

In practice, we cannot test the quadratic relationship for every site in the compiled experimental dataset (*CO₂_exp*) because of the limited number of N addition levels used at most experimental sites – at least five data points may be needed to determine a quadratic curve with an acceptable level of confidence. For this sake, we conducted an additional analysis using data from experimental sites with at least five N input levels (including the control). For sites where multiple N addition experiments were conducted, data from experiments of the same duration were considered together, while data from experiments of different durations were considered separately. Four sites met

the criteria (i.e., having more than 4 levels of N addition), and significant (or nearly significant) quadratic relationships were found at all sites (Supp. Fig. S4; attached below).

Fig. S4 Relationship between N input rate and the N-induced change in soil respiration rate (ΔSR_N) in four experimental forests with at least five N addition levels (including the control). In each panel, a gray point represents one N addition level and the corresponding ΔSR_N . Here, ΔSR_N was calculated as the difference in SR in paired control and experimental N addition plots. The blue curves show the quadratic models fitted to the experimental data of each forest. Blue fonts show the goodness of fit of the quadratic models. Black fonts show the basic information for each experimental forest. The information was retrieved from the original sources of data: Forest A (Forsmark et al. 2020), Forest B (Yu 2019, Zhao 2019), Forest C (Yu et al. 2019), Forest D (Geng et al. 2017, Peng et al. 2021).

In addition to Fig. 1, we provided a Supp. Text S1 and Fig. S1 (attached below) which contained information about the theoretical basis for the hypothesized biphasic and abrupt responses. We also revised the introduction section to provide additional content about mechanisms underlying the biphasic and abrupt collapse hypotheses (see L92–104).

Fig. S1 Theoretical basis and graphical illustration of the overall response factor (f_a) and relative response factor (f_r) of soil respiration to N input. N_{dep} is the atmospheric N deposition rate. SR_{CL} is the soil respiration rate in the control plot receiving background N deposition only. SR_{+N} is the soil respiration rate observed in the experimental plot receiving artificial N addition. N_{add} is the N addition rate.

#4. 2. It is unclear why the authors chose to analyze positive and negative SR responses separately. SR response to N deposition is a continuous variable that can range from positive to negative within the same experiment. While I agree that different pathways can drive either positive or negative responses, this does not necessarily mean that these responses should be analyzed in isolation. It is more likely that both positive and negative mechanisms operate simultaneously, with the net response depending on which mechanisms dominate under a given N deposition rate and background conditions. Without a clear theoretical justification for this separation, this analytical approach seems somewhat arbitrary.

Response: This is a very good question. In fact, we did not separate the positive and negative SR responses in our initial analysis, hypothesizing that low and short-term N enrichment would increase SR, whereas high or long-term N enrichment would decrease SR (which is essentially what N saturation theory would predict)(Aber et al. 1998). However, the patterns revealed (see fig. 1 below) suggest that low N additions could increase SR (and its components, RR and MR) in both N-limited and N-saturated forests. This implies that the initial hypothesis about the response of SR to N input was incomplete.

fig. 1 Local polynomial regression (LOESS) models of N input rate and the N-induced changes in soil respiration (ΔSR_N), root respiration (ΔRR_N), and microbial respiration (ΔMR_N). Different panels show experimental N addition data from N-limited forests, N-saturated forests, and long-term experimental forests. In each panel, gray points show observations of ΔSR_N (or ΔMR_N or ΔRR_N) derived from the *CO₂_exp* dataset; each point represents a site-year-N addition level combination. The red curve shows the LOESS model, and the pink shading shows the range of uncertainty of the model.

Therefore, we revised our initial hypothesis based on biochemical theories and mechanisms underlying soil respiration, including the metabolism theory (Brown et al. 2004) and stress-induced community shifts (Egidi et al. 2023). Considering the theoretical and observational connections between belowground living biomass and soil respiration changes (Brown et al. 2004, Treseder 2008, Xing et al. 2022b), the increased and decreased SR by N addition may arise from different mechanisms functioning at different scales – the increased SR results from the biphasic effect of N on organisms (i.e., the low N-facilitated biomass growth and high N becomes toxic), whereas the decreased SR results from the community-level collapse (i.e., N-induced biodiversity loss and the mortality of living biomass).

When analyzing the increased and decreased SR by N input separately, the biphasic (quadratic) responses of increased SR by N input could be seen in almost all forests (see red curves in fig. 2 below). The abrupt collapse of the N-induced decrease in SR (blue curves) is not as pronounced as in Fig. 3, probably because of the different initial N availability and basal soil respiration rate across the sites. It should be noted that the increased and decreased SR change divergently with increasing N input, suggesting that the two mechanisms operate in parallel. This is exactly the reason why we need to analyze the positive and negative SR responses separately, only by which we could quantitatively reveal the SR response curves and predict the N deposition effects on forest SR. We added information in the main text (L132–135), Supp. Text S1, and Fig. S1 to justify the separation of the increased and decreased SR.

fig. 2 Local polynomial regression (LOESS) models of N input rate and the N-induced increase and decrease in soil respiration (ΔSR_N), root respiration (ΔRR_N), and microbial respiration (ΔMR_N). Different panels show experimental N addition data from N-limited forests, N-saturated forests, and long-term experimental forests. In each panel, gray points show observations of ΔSR_N (or ΔMR_N or ΔRR_N) derived from the *CO₂_exp* dataset; each point represents a site-year-N addition level combination. The red curve shows the LOESS model for the N-induced increase of SR (or RR or MR), and the blue curve shows the LOESS model for the N-induced decrease of SR (or RR or MR). The

colored shadings show the ranges of uncertainty of the models.

On a regional scale, both positive and negative responses are likely to exist at the same time (but in different forests). In a given forest ecosystem, the biotic community would either be in a collapse-rebuild phase (with decreased SR) or still benefiting from N supply (with increased SR). In fact, Zheng et al. (2022) provided a nice analysis of such phased changes using data from a long-term experimental N addition site. However, such long-term experimental data capturing the different phases of SR change are scarce. Therefore, it has been challenging to analyze the positive and negative responses combinedly, because the same N addition level could lead to different SR responses depending on which phase the forest is in - this may be an important reason why previous research that did not separate the different responses was unable to reach a universal framework for explaining N-induced SR changes.

#5. 3. Lines 526–535:The distinction between “N-limited,” “N-saturated,” and “long-term” experimental forests is not clearly justified. In particular, the rationale for defining three years as the threshold for high N stress is unclear—why not two, five, or ten years? Moreover, N stress is not solely determined by duration; the N addition rate also plays a critical role. For example, a low N addition rate over a longer period (e.g., 10 kg ha⁻¹ yr⁻¹ for four years) may impose less stress than a high N addition rate over a shorter period (e.g., 100 kg ha⁻¹ yr⁻¹ for three years), given differences in total N input. This issue requires further clarification and justification.

Response: Considering the potential influence of background N status (i.e., N-limited vs. N-saturated) and duration of N addition on the SR responses, we separately analyzed experimental data from N-limited forests, N-saturated forests, and long-term experimental forests. Please refer to the response to Comment #8 for a detailed explanation of the distinction between N-limited and N-saturated forests in this study.

Following previous studies (Wang and Tang 2019, Xu et al. 2021, Wang et al. 2023), we distinguished between forests where experiments were conducted for less than or more than 3 years. Experiments of two years were barely considered as long-term experiments in any study.

A realistic reason for the adopted experimental duration threshold (i.e., 3 years) is that many of the experimental data were collected through master’s or Ph.D. projects, and the length of these experiments is typically no more than 3 years. Experiments longer than 3 years are few, which means that not many observations would be left in the long-term experimental data set and a high uncertainty in the derived response factor of SR to N addition would occur, if we set the threshold of long-term experiments to a higher level, such as 5 years.

In order to fully address the reviewer’s concern, we tried reanalyzing the data using different thresholds (5 yrs, 10 yrs). Other thresholds could be tried using the supplementary data and R code (downloadable from <https://data.mendeley.com/preview/wfdfjds6by?a=c296db55-569b-4c5f-8b65->

d85e9e51dd3a). As can be seen in fig. 3 below, the response patterns remain unchanged, despite high uncertainty in the responses of SR in long-term experimental forests which is likely due to the limited number of observations classified as long-term experiments. Therefore, we prefer to keep using 3 years as the threshold. We added some text in L436–437 to justify the used threshold.

fig. 3 Using different thresholds for analyzing increased soil respiration (SR) by N input in N-limited, N-saturated, and long-term experimental forests (supplemental to Supp. Fig. S3). (A–C) Piecewise linear regression models relating N input rate to the response factor of increased SR by N input (f_a) in experimental forests. Gray points show the observations of f_a derived from experimental dataset (CO_2_exp). Red (gray) lines and font show the fitted linear models for f_a and low (high) N input rate. The linear models for f_a and low N input rate (red font) were then used to infer the quadratic models of increased SR (ΔSR_{N-inc}) and N input. (D–F) Quadratic models on the increased soil respiration (ΔSR_{N-inc}) and N input rate in forests. Gray points show the observed ΔSR_{N-inc} in experimental forests. Black points show the moving average of ΔSR_{N-inc} for every 10 kgN ha⁻¹ yr⁻¹; this aggregation of data was to mitigate the influence of extreme values. The red lines show the quadratic models calculated from the linear models. Pink dashed lines show the range of uncertainties of the quadratic models. Red font shows the correlation coefficient (R) between the observed and predicted mean ΔSR_{N-inc} . Blue lines and font show the turning points of the quadratic models. “N-limited” forests refer to experimental forests which were initially N-limited and where N addition experiments had been conducted for no more than 3/5/10 years; “N-saturated” forests refer to experimental forests which were initially N-saturated and where N addition experiments had been conducted for no more than 3/5/10 years; “long-term” forests refer to those where N addition experiments had been conducted for more than 3/5/10 years, regardless of the initial N status.

Also, we considered following the practice of some studies (such as Chen and Chen (2023)) and analyzing the response of SR to total N addition over the entire experimental period. But still, a N addition of 100 kg ha⁻¹ yr⁻¹ for one year may have

different ecological effects than a N addition of 25 kg ha⁻¹ yr⁻¹ for four years – the total N addition may be the same but the N losses through leaching and gaseous emission could be very different, leading to different availability of retained N. Therefore, we took extra efforts to divide the original data into short-term (which was further divided into N-limited and N-saturated forest subsets) and long-term experimental datasets, and then analyzed the relationship between N addition rate and SR for each subset, intending to separate the effects of N addition rate, duration, and background N status on forest SR.

#6. 4. For the data derived from N addition experiments, it is essential to clarify how potential non-independence among data points within each original study was accounted for in the statistical analyses. Addressing this concern would help ensure the robustness of the findings.

Response: In our analysis, to avoid the over-representation of data from the same experiment (i.e., data from the same plot-year may have been analyzed and reported in different papers), we summarized data from the same year and N addition level by taking their averages. Each unique site-year-N addition level combination was used as a data point when we calculated the response factors of SR to N addition.

Nonetheless, non-independence of the calculated response factors (as effect size metrics) could result from: (1) The shared control was used to calculate the overall response factors of SR to different N additions applied at the same site, so that the derived overall response factors could be correlated (*Cause 1*). (2) The effect of N addition was estimated for each experimental year so that, for a given site, the response factors corresponding to different years could be temporally correlated (*Cause 2*).

In this study, we used response factors as metrics of effect size rather than the commonly used response ratio (see response to Comment #19 for details). The response factors were used to derive a biphasic/abrupt-transition relationship between SR and N inputs, instead of deriving an overall effect size metric showing whether N inputs increase or decrease SR. Although these differences between our study and common meta-analysis were critical for revealing the complex, phased relationships between SR and N inputs, they prevented us from directly testing the influence of non-independence of data points on the revealed relationships using a meta-regression method as suggested by Gleser and Olkin (2009).

However, the influence of data point non-independence could generally (and somewhat indirectly) be addressed by using an alternative effect size metric or by performing the analysis on subgroups of the dataset (Noble et al. 2017). To account for the non-independence arising from *Cause 1*, we have defined and calculated an alternative effect size metric, local response factor (see Supp. Text S2 for details), to verify the revealed phased relationship between SR and N inputs. The local response factor was calculated using data from any two N addition treatments within a given range of N input rates. That is, the local response factors do not have a “shared control” problem like the overall response factors. Because the local response factors showed the same pattern as the overall response factors in how SR responded to increasing N inputs (Fig.

7, Supp. Fig. S6), *Cause 1* of non-independence is unlikely to challenge our finding about the response of SR to N inputs.

Regarding non-independence due to *Cause 2*, we have tested and validated that the revealed patterns persist regardless of using subsets of the experimental data from short-term or long-term experiments (i.e., data from experiments within/after than three years; Supp. Figs. S3, S5). The fig. 3 in the response to Comment #5 may further demonstrate that the patterns revealed are independent of the temporal range of the experimental data used for analysis, meaning that the potential temporal correlation of data points is unlikely to challenge our finding about the response of SR to N inputs.

Additionally, the same patterns revealed in our global analysis (Fig. 3) and site-level analysis (Supp. Fig. S4) may indirectly support that the revealed the response of SR to N inputs is unlikely to be challenged by the potential spatial correlation of the experimental sites. Our analysis on the observational dataset (Fig. 4), which is independent of the analysis on the experimental dataset, also revealed the same patterns. We believe that all these evidences can support the robustness of our findings. We revised the main text (L421–423) and added a Supp. Text S3 to clarify.

#7. 5. Use of Abbreviations: The manuscript introduces and relies on numerous abbreviations, many of which are not clearly defined, making the text difficult to follow. In several cases, I only fully understood their meaning after referring to the Methods section. I recommend using abbreviations only when necessary and ensuring that all abbreviations are explicitly defined in the introduction to improve readability.

Response: We have added a glossary of terms and abbreviations at the beginning of the paper (Box 1). Unnecessary abbreviations have been avoided wherever possible. Hopefully the revised manuscript is more readable now.

Reviewer #2 (Remarks to the Author):

Reviewer Comments to the Author:

#8. Cen et al. systematically compiled and analyzed a dataset of annual soil respiration fluxes observed in forests worldwide where nitrogen addition experiments were conducted. Meanwhile, they applied a natural gradient approach to analyze the SRDB database and the global nitrogen deposition dataset. Based on these analyses, they proposed a general framework for the effects of nitrogen deposition on soil respiration in global forests. Subsequently, they assessed the net effect of nitrogen deposition on the global forest soil respiration budget. The proposed general framework for the response of soil respiration to nitrogen inputs is innovative. In addition, they have also made several new estimations of large-scale indicators. I have a few suggestions for your consideration.

First, it is important to note that N addition typically promotes soil respiration under conditions where nitrogen saturation has not yet been reached. If a forest is already in a nitrogen-saturated state, the more plausible outcome would be nitrogen addition primarily leading to inhibition of soil respiration. In most ecosystems, nitrogen addition initially stimulates soil respiration, followed by a suppression phase, and eventually leads to a collapse, resulting in a new equilibrium. Once this equilibrium is reached, the ecosystem becomes nitrogen-unsaturated, and this cycle may repeat. I recommend that the authors clarify this point in the context of nitrogen saturation and its role in the nitrogen addition experiments discussed in the manuscript.

Response: This manuscript was intended to clarify the effects of N addition on SR in N-limited and N-saturated forests. As the traditional N saturation theory would predict, N addition increases SR in N-limited forests and decreases SR in N-saturated forests (Hypothesis 1). However, if we consider the adaptation of biotic community to environmental stress, such as that caused by high anthropogenic N deposition, it is unlikely that excessive N could persistently suppress biotic activities and respiration at the community level – the community would adapt to high N availability through trait plasticity or species composition change, thereby further making use of the additional N supply (Hypothesis 2).

In a previous study (Cen et al. 2025), we determined the N limitation or saturation status of global forests using the sensitivity of soil N₂O emission to N deposition (s_N) as an indicator. In previous papers, researchers reported the N limitation or saturation status of forest sites based on either the forest growth response to N addition, or the N leaching level. Those observations were used to determine the s_N threshold between N-limited and N-saturated forests, based on which we classified global forests to be N-limited or N-saturated.

Using the derived N status map of global forests (Fig. 2)(Cen et al. 2025), we were able to separately analyze the SR responses to N addition in N-limited and N-saturated forests. It was found that low N addition increased SR in both N-limited and N-saturated forests (see figs. 1 and 2 in the response to Comment #4). This finding is partly different from what the traditional N saturation theory would predict (i.e., Hypothesis 1), but explainable if we incorporate the community structure change mechanism (Hypothesis 2) – at least the microbial communities could quickly change and make use of the high N availability for biotic activities, and hence SR could further increase even when plants are barely responsive to N addition (which is a sign of N saturation).

There are some ongoing controversies about the definitions of N limitation and N saturation, and we agree it is perhaps reasonable to think that the community becomes N-limited again after community reconstruction, considering that the newly recruited copiotrophs could keep growing and respond positively to further N addition. However, the background N availability and N tolerance level of organisms in the reconstructed community is for sure different from the initial, N-limited community. For better distinguishing between the old and new communities, we hesitate to call the reconstructed community “N-limited”. For this sake, the community change was intentionally not drawn as a closed cycle in Fig. 1 – it would likely be a spiral rather than a cycle if going to further stages after community reconstruction, although we

didn't go that far to validate the entire spiral model.

It is perhaps also controversial to call the rebuilt community N-saturated, but we haven't thought of other terms that could provide readers with a straightforward impression about the different N status of communities. In the context of this study and previous studies of ours, the N-limited and N-saturated forests are intended to refer to two types of forests with different N availability that exhibit disparate responses to future N deposition and climate change (see Box 1 below from Cen et al. (2025)).

We have clarified the definition of N-limited/N-saturated forests in the glossary of terms at the beginning of the paper. We have also added some text in the Discussion section to clarify the similarities and differences in SR responses in N-limited and N-saturated forests (L251–264).

Box 1. N status and N availability

In the context of this study, the term N status is used to refer to either N limitation or N saturation. When N supply exceeds biotic N demand, forests are considered to be in an N-saturated state; otherwise, forests are N limited. Given the inherent challenges in quantifying N supply and biotic N demand at the ecosystem level, researchers have used various indicators (see Note S1) to determine forest N status. A common misconception about N saturation is that once a forest becomes N saturated, the N pool no longer increases. Actually, N-saturated forests could still transform and fix additional N, but not as efficiently as before N saturation. N saturation leads to a large fraction of the N supply being lost, either via gaseous emissions or leaching. Using the sensitivity of soil N₂O emissions (a form of gaseous N loss) to N deposition as an indicator, we classified global forests into two categories (N limited/ N saturated), although there may be an intermediate state where forests are in transition between N limitation and N saturation. From a practical standpoint, the N-limited and N-saturated forests in this study could also be understood as two types of forests with different N availability that exhibit disparate responses to future N deposition and climate change. In this paper, the term N availability is used to refer to N supply relative to biotic N demand, as defined by Mason et al.²⁸ The continuous change of N availability (caused by, for example, increased N deposition, enhanced atmospheric CO₂ levels, harvesting) may lead to a shift in forest N status (Figure X1).

Figure X1. Graphical display of N availability and N status changes

#9. Second, in multi-level N addition single-point experiments, if the data can be fitted to a unimodal curve, it would be helpful to assess whether the majority of experimental fits exhibit a unimodal pattern. Such evidence could help reduce the noise caused by heterogeneity between experiments, thereby strengthening the reliability of the conclusions drawn in the manuscript. Although the dataset is relatively small, I recommend that the authors attempt to provide more substantial evidence to support their findings, which would enhance the robustness of the study and its conclusions.

Response: Thank you for this great suggestion. We used data from four experimental sites having more than four N addition levels and revealed significant quadratic relationships (Supp. Fig. S4; attached below), supporting the overall biphasic response of SR to N input. The supplementary R code file has been updated to reflect how we filtered out the sites having more than four N addition levels and how Fig. S4 was produced.

Fig. S4 Relationship between N input rate and the N-induced change in soil respiration rate (ΔSR_N) in four experimental forests with at least five N input levels (including the control). In each panel, a gray point represents one N addition level and the corresponding ΔSR_N . Here, ΔSR_N was calculated as the difference in SR in paired control and experimental N addition plots. The blue curves show the quadratic models fitted to the experimental data of each forest. Blue fonts show the goodness of fit of the quadratic models. Black fonts show the basic information for each experimental forest. The information was retrieved from the original sources of data: Forest A (Forsmark et al. 2020), Forest B (Yu 2019, Zhao 2019), Forest C (Yu et al. 2019), Forest D (Geng et al. 2017, Peng et al. 2021).

#10. Finally, this study involves numerous estimations and predictions, making the methods for model construction and the reliability of predictions crucial. I recommend that the authors provide more detailed and reliable explanations, as outlined in the Specific Comments section below.

Response: We revised the Methods section accordingly to provide more details about the models and predictions. Please refer to the responses to Comments #29–34.

Specific Comments:

#11. Line 53-55: "High exogenous N inputs (deposition or fertilization) "and" long-term N additions" are somewhat repetitive." You should clarify what you want to express.

Response: Because there are multiple comments about the use of "N stress", the term has been deprecated in the revised manuscript. The explanation to the term (originally in L53–55) was therefore removed.

#12. Line 56-57: In my opinion, nitrogen availability refers only to the amount of nitrogen in an ecosystem that is available for biological use.

Response: We agree that traditionally, N availability refers to the amount of biologically available N. This definition of N availability (i.e., N supply in relative to biotic N demand) follows that of Mason et al. (2022). We added the reference here to clarify (Box 1 in the main text).

In this paper, the term N availability was used in parallel with N status (N limitation/saturation) to indicate that the continuous change of N availability caused by environmental changes can lead to alteration of ecosystem N status, thereby influencing the SR response to N input. Please also refer to Box 1 in the response to Comment #8 for details.

#13. Line 60-61: "background N stress" It should be the availability of nitrogen, not every forest is under N stress.

Response: This sentence has been removed, but we have deprecated the use of "N stress" throughout the manuscript.

#14. Line 66: "Soil respiration (SR)" Rs is more commonly used as the abbreviation for soil respiration.

Response: Yes, we initially used Rs as the abbreviation for soil respiration. However, since we also need an abbreviation for the N-induced change of soil respiration, we realized that RS_N looks somewhat confusing (the lowercase "s" and subscript "N" may look like a connected "SN"). Therefore, we ended up using SR and SR_N as abbreviations of soil respiration and the N-induced change of soil respiration.

#15. Line 69: "mixed effects" diverse effects

Response: Revised accordingly (L44).

#16. Line 112-114: The studies by Janssens, I. A. et al. and Zhou, L. Y. et al. are both

based on meta-analysis results, revealing the overall effects of numerous studies, many of which show that nitrogen addition promotes soil respiration. However, the entry point you mentioned, that low nitrogen promotes soil respiration, seems inconsistent with their findings, and this point comes across as somewhat confusing.

Response: Previous meta-analyses of experimental N addition data revealed that N deposition reduced global forest SR budget. Some recent meta-analyses, however, suggested that realistically low N deposition increased global forest SR budget. The inconsistency of their conclusions may result from the potential bias in the available experimental N addition data including: (1) the N addition level used in most N addition experiments are higher than the real-world N deposition level; (2) the high representation of N-saturated forests (e.g., temperate forests close to human settlements) in global N addition experiments; (3) the mixed use of short-term and long-term experimental data. Also, the meta-analysis method itself may have limited capability to quantitatively reveal the complex, phased relationship between N input and SR – please see the response to Comment #19 for details.

For these reasons, we conducted this research, separately and quantitatively accounted for the effects of N input rate, background N status, and N addition duration on SR. We briefly mentioned the different effects of N deposition on SR found in different meta-analyses (L78–89) – we were afraid that too much information about methodology here in the Introduction section would distract readers. Instead, we added some discussion (L345–364) to clarify the reasons behind the different arguments about how low N inputs influence SR.

#17. Line 134-136: The current phrasing may give the impression that the biphasic dose-response relationship is solely due to soil acidification. It is recommended to separate the explanation and the summary for better clarity.

Response: Revised accordingly. Please see L100–104.

#18. Line 187-189: Rs, Ra, and Rh should be the commonly used abbreviations for soil respiration and its components.

Response: Yes. For the same reason as specified in the response to Comment #14, we ended up using SR, RR, and MR to stand for soil respiration and its components.

#19. In addition, I have a question: The effect size of such an abrupt transition is often disproportionate to the N addition level triggering the change. How does the algorithm used in this paper differ from the response ratio commonly used in meta-analysis? This is just a curiosity and not a challenge to your calculation method.

Response: This is a good question. In meta-analysis, the response ratio (RR) is commonly quantified as the target variable measured in the experimental plot (X_{EXP}) relative to the background value (measured in the control plot; X_{CL}). That is, $RR = X_{\text{EXP}} / X_{\text{CL}}$. Response ratio can measure the qualitative effect of the experimentally changed

environmental condition (i.e., whether a target variable is increased or decreased by N deposition, temperature, precipitation, or atmospheric CO₂ level, etc.). However, this metric can hardly be applied to reveal a complex, phased relationship such as that between N input and soil respiration. This might be a reason why previous meta-analyses (which used response ratio for measuring the N input effects on SR) led to different conclusions whether atmospheric N deposition increases or decreases forest soil respiration on a global level.

In this study, based on biochemical and ecological theories, we used different metrics for quantifying the N deposition effects on soil respiration. Due to the positive relationship between biomass and respiration (“power law”)(Reich et al. 2006), and between enzyme concentration and decomposition (“Michaelis-Menten kinetics”)(Michaelis and Menten 1913), it is inferred that there is an intrinsic relationship between the amount of N added and the increased respiration (Supp. Fig S1, attached below). Therefore, we defined an “overall response factor” (f_a ; calculated as the increased soil respiration per unit of artificial N addition) to reveal the dose-response relationship between N input and the increased soil respiration.

Niche complementarity allowed the increasing stress from high N availability to be shared among species, ensuring the stability of the community and preventing a gradual decline of biodiversity and living biomass. However, excessively high N availability could at a point exceed the N tolerance limit of the entire community and kill the N-sensitive oligotrophs, leading to abrupt reductions in the living biomass and respiration of the community (Bobbink et al. 2010, Zheng et al. 2022, Egidi et al. 2023). The effect size of such an abrupt transition is often disproportionate to the N addition level triggering the change. Therefore, we defined another “relative response factor” (f_r) to quantify the magnitude of the reduction, calculated as the decreased soil respiration relative to the initial soil respiration (Supp. Fig. S1).

Details of the response factors have been provided in Supp. Text S1.

Fig. S1 Theoretical basis and graphical illustration of the overall response factor (f_a) and relative response factor (f_r) of soil respiration to N input. N_{dep} is the atmospheric N deposition rate. SR_{CL} is the soil respiration rate in the control plot receiving background N deposition only. SR_{+N} is the soil respiration rate observed in the experimental plot

receiving artificial N addition. N_{add} is the N addition rate.

#20. Line 240, 248: The change in soil respiration (ΔSRN) should be explained in more detail, as I am unclear about how the change is calculated up to this point.

Response: Revised accordingly. Please see L845–846 and L865–866 for the revision.

#21. Line 265-266: The classification of "N-limited forests," "N-saturated forests," and "long-term experimental forests" is confusing, as it is unclear why "long-term experimental forests" should be considered separately.

Response: Previous research shows that the SR response to N inputs may depend on the initial N status (i.e., N-limited or N-saturated), the rate of N addition, and the duration of N addition (Janssens et al. 2010, Chen and Chen 2023, Liu et al. 2023). To distinguish between the influence of different factors, we divided the original data into short-term (which was further divided into N-limited and N-saturated forest subsets) and long-term experimental data (which was not further divided because of the relatively few observations from long-term experiments; see response to Comment #5), and then analyzed the relationship between N addition rate and SR for each subset. Data from N-limited, N-saturated, and long-term experimental forests showed clear differences in their SR responses (see Fig. 5 in the main text), which may support the validity of separately analyzing the datasets. We have added some text here to clarify the reason for the classification (L189–196).

From another perspective, previous research that did not distinguish between short-term and long-term experimental data may overestimate the negative effect of N deposition on SR, because in long-term N addition experiments, plant and microbial properties could be altered by the accumulated high N availability, causing forest SR to deviate from the natural response to N deposition. We therefore advocated that the short-term and long-term experimental data should be used to reveal the short-term response and long-term adaptation of SR to changes in N deposition, respectively (L361–364).

#22. Line 326-327: I have been unable to understand the meaning of P_{inh} in Fig. 7. According to this framework, the results from the N-limited and N-saturated forest datasets are the most important evidence and should be highlighted in the main text.

Response: Sorry for the mistake here. P_{inh} was the previously used abbreviation for the probability of SR being inhibited by N addition. P_{dec} was used instead of P_{inh} in a revision, but I forgot to update Fig. 7.

According to the results (Supp. Figs. S3, S5, and Fig. 4), SR in N-limited and N-saturated forests showed similar patterns in response to increasing N inputs. The major difference between them was that for the same level of N input, SR is more likely to decrease in N-saturated than in N-limited forests (Fig. 5). Both the inter-biome differences in SR responses and the inter-N-status differences may be of interest to readers. To avoid making Fig. 3 look too busy, we have provided some results from N-

limited/N-saturated forests as Supp. Figs. S3 and S5. Nevertheless, we have highlighted the inter-N-status differences in Fig. 4, Table 1, and also in the Discussion section (L251–264).

#23. Line 353: In N-limited forests, the old communities transition into new communities. Does this mean that the new communities represent N-saturated forests? Will their nitrogen demand increase?

In Appendix Text S2, with increasing N input, the local response factor (f_b) of SR to N input shifted from positive to negative and then back to positive in N-limited forests across all biomes (Fig. S5).

Response: Both experimental and observational data (Fig. S6 and Fig. 4) suggest that the newly recruited copiotrophs in the reconstructed new communities could further utilize the additional N for biological activities, thereby further increasing soil respiration. In a sense that the new communities have a higher N-tolerance level and higher N-availability than the original N-limited communities, we considered it to be N-saturated. On the other hand, we can see the reasonableness of considering the reconstructed communities to be N-limited, as the newly recruited copiotrophs could utilize the additional N (although we don't think the biological N demand increase, but that oligotroph species were replaced by copiotrophs).

This is more a question of how we define N limitation and N saturation. In the context of this study, the N-limited and N-saturated forests are intended to refer to two types of forests with different N availability that exhibit disparate responses to future N deposition and climate change. Please refer to the response to Comment #8 for more details.

#24. Line 390: The two explanations for nitrogen addition inhibiting soil respiration should not be seen as an incremental relationship like "even".

Response: Revised accordingly. Please see L282–284 for the revision.

#25. Line 417-419: The theme of this paragraph is to discuss the similarities and differences across different ecosystems and nitrogen status categories. The mention of response range here seems somewhat out of place.

Response: Revised accordingly. Please see L305–319 for the revision.

#26. Line 422: The explanation, i.e., from N-limited to N-saturated to long-term experimental forests, might be more helpful to the reader if mentioned earlier in the text.

Response: Revised accordingly. We added explanation in the Results section about the classification of experimental forests (L189–196).

#27. Line 451-455: This content should be placed at the end in the conclusion and significance section. It might serve as a transition to the next section, but it feels somewhat abrupt when read.

Response: Revised accordingly. Please see L345–348 for the revision.

#28. Line 451-482: It is recommended to organize and categorize these insights and limitations. As it stands, the reading feels somewhat disorganized and chaotic.

Response: Revised accordingly (L349–364). Hopefully it reads more organized now.

Lines 499-505: Did you use only the SRDB data, or did you also include the CK data from your own CO₂_exp dataset?

Response: The SRDB data (from non-experimental forests) were used to build the random forest model and derive the soil respiration budget of global forests. CO₂_exp dataset was only used for quantifying the N input effects on SR.

Although the CK data from CO₂_exp may as well be used in combination with SRDB data for building the random forest regression model, we don't think the model would be significantly changed considering the relatively small size of CO₂_exp (with 390 obs. from CK plots) as compared to the SRDB data (n = 3689).

#29. Lines 608-609: What was the basis for selecting the predictor variables? Why were these specific variables ultimately chosen?

Response: The predictors were selected based on (1) the availability of data for each experimental site and for global forests; (2) the mechanistic connection between SR and climate (Bond-Lamberty and Thomson 2010, Huang et al. 2020), N deposition (Reay et al. 2008), and soil texture (Wang et al. 2003).

To improve the performance of the regression model for prediction, we further incorporated soil pH, soil organic carbon, and soil nitrogen content variables. These variables were also found to be related to soil respiration in previous research (Chen et al. 2014, Bae et al. 2015).

#30. Lines 600: The selection and determination of random forest model parameters (e.g., Ntree, Mtry, and nodesize) have a significant impact on model performance. The authors should clarify how this issue was addressed in the Methods section.

Response: Thank you for pointing this out. Mtry parameter (i.e., number of variables tried at each split) was optimized using “tuneRF” function in “randomForest” package. In many cases, the out-of-bag error rate was the lowest when Mtry was set to 4 (fig. 4). This also agrees with the rule of thumb (i.e., optimal Mtry being approximately the

square root of the number of predictors). We have clarified the parameter selection in the Methods section (L537–539) and in Supp. Tables S2, S4.

The error rate barely changed when Ntree (i.e., number of trees) was above 200. To keep a balance between model error rate and computational cost, we set the Ntree parameter to be 500.

fig. 4 Out-of-bag error of random forest model when the Mtry (left panel) and Ntree (right panel) parameters were set to different values.

The model performance did not change significantly when we set the nodesize to be 1, 5, or 10 (see below for the model output). Therefore, we kept using the default setting of the randomForest function (i.e., nodesize equals 5 when the response variable is a continuous variable).

```
Call:
randomForest(formula = Rs ~ MAT + MAP + MAT.cv + MAP.cv + MAN + MAN.cv + Clay + Sand + SOC + TN + pH + sN, data = rf.data, nodesize = 1, mtry = 4, ntree = 500, importance = TRUE, proximity = TRUE)
Type of random forest: regression
Number of trees: 500
No. of variables tried at each split: 4

Mean of squared residuals: 111496131
% Var explained: 51.77

Call:
randomForest(formula = Rs ~ MAT + MAP + MAT.cv + MAP.cv + MAN + MAN.cv + Clay + Sand + SOC + TN + pH + sN, data = rf.data, nodesize = 5, mtry = 4, ntree = 500, importance = TRUE, proximity = TRUE)
Type of random forest: regression
Number of trees: 500
No. of variables tried at each split: 4

Mean of squared residuals: 112353284
% Var explained: 51.39

Call:
randomForest(formula = Rs ~ MAT + MAP + MAT.cv + MAP.cv + MAN + MAN.cv + Clay + Sand + SOC + TN + pH + sN, data = rf.data, nodesize = 10, mtry = 4, ntree = 500, importance = TRUE, proximity = TRUE)
Type of random forest: regression
Number of trees: 500
```

No. of variables tried at each split: 4

Mean of squared residuals: 113407470

% Var explained: 50.94

#31. The soil respiration data were likely collected over different time periods, leading to potential inconsistencies. How did the authors account for the influence of study duration in their analysis?

Response: The duration of experimental N additions influences soil respiration responses. Therefore, we separately analyzed soil respiration data from experiments that lasted less than or more than three years (Xu et al. 2021, Wang et al. 2023). The interannual variation of soil respiration was less of a problem because the analysis was performed using data from paired control and N addition plots. The difference in SR between the plots was therefore attributable to N input rather than the temporally varying climatic factors.

On the other hand, the interannual variation of soil respiration may add to the uncertainty in the estimated global forest SR budget, because we did not distinguish between data collected from different years. The derived budget should thus be interpreted as the mean SR budget of the past few decades (when soil respiration rates were measured in the field). However, because the interannual variation of SR is relatively small as compared to the basal value (Huang et al. 2020), the estimated SR budget and spatial pattern are comparable to those reported in previous studies (Raich and Schlesinger 1992, Wang et al. 2010, Zhao et al. 2024)(Supp. Fig. S7).

#32. The study by Ni Huang, Spatial and temporal variations in global soil respiration and their relationships with climate and land cover, has also conducted global-scale soil respiration predictions, achieving an R^2 of 0.6–0.8. In contrast, the R^2 in this study is only 0.25. How reliable are the predictions? Have the authors considered alternative machine learning approaches, such as support vector machines (SVM), to improve model performance?

Response: Thank you for the suggestion. Our original model for predicting SR had an R^2 of 0.45. To improve the performance of the model, we incorporated soil chemical variables (i.e., soil pH, organic carbon content, and soil nitrogen content) which were mechanistically related to SR (Chen et al. 2014, Bae et al. 2015). The new model has an R^2 of 0.51 (Supp. Table S4). Although the model performance may be further improved if we build plant-functional-type-specific models as Huang et al. (2020) did, we hesitate to make the model such complicated because the major focus of this study is to reveal the N deposition effects on SR. The random forest model for predicting SR serves to derive the basal SR rates, on which basis we could quantify the contribution of N deposition to the current SR budget – the model barely influenced our results about the N deposition effects. Also, the global forest SR budget we estimated is comparable to those reported in previous studies (see Supp. Fig. S7, attached below).

Fig. S7 Comparing the estimated global forest soil respiration budgets from previous studies (Raich and Schlesinger 1992, Wang et al. 2010, Zhao et al. 2024) with the results of this study.

The random forest model for the relative response rate of decreased soil respiration (f_r) had an R² of 0.25 (original Supp. Table S1). It should be noted, however, that this model was only used to analyze the partial dependence of f_r on N input rate, not to predict f_r . We also included soil chemical variables to improve model performance (current R² is 0.34; Supp. Table S2). The derived partial dependence plots barely changed before and after the model revision, because the relationship depends mostly on f_r and N input rate, and little on other variables used in the model (Fig. 3).

#33. Lines 651-658: What was the basis for selecting this nonlinear regression model? Is it the optimal model? The authors should provide a more detailed explanation, as the current description remains unclear.

Response: Pdec is a probability with lower and upper bounds of 0 and 1. We tried using several different types of models that fit for such a variable, including the Weibull models (W1.2 and W2.2), the log-logistic model (LL.2), and the Michaelis-Menten model (MM.3). Since the MM.3 model has the lowest AIC value (see model comparison below), it was considered the optimal model and was used for the non-linear regression analysis.

Model fitting and selection were performed in R using the “drc” package (Ritz et al. 2016). The supplementary R code has been updated to reflect the model selection process. We also added some text in the Methods section (L565–570) and a Supp. Table

S6 to clarify.

```
> ar.all <- drm(P_inh ~ N_in, data = subset(output, N_in <= 400), Nstat, fct = MM.3(fixed = c(0, 1, N
A)))
> mselect(ar.all, list(LL.2(), W1.2(), W2.2()), icfct = AIC)
  logLik      IC Lack of fit  Res var
W1.2 33.79337 -53.58674      NA 0.04859006
LL.2 33.39288 -52.78577      NA 0.04870814
W2.2 32.34107 -50.68213      NA 0.04901963
MM.3 19.23693 -30.47386      NA 0.05258460
```

#34. Using this empirical equation for prediction does not seem to be theoretically superior to machine learning. Why not use machine learning to predict Pdec for any N input rates?

Response: Machine learning algorithms often work better with large datasets (e.g., with thousands of obs.). Because of the relatively few Pdec data available (n = 330), machine learning algorithms may not fit for predicting Pdec.

Also, an important purpose here is to explore how Pdec changes with N input rates and across forests in different N status. The “black box” nature of machine learning algorithms may lower the model interpretability (Pichler and Hartig 2023), making it difficult to convince readers of a solid, consistent relationship between Pdec and N input (or N status) like what was shown in Fig. 5. Therefore, we prefer to keep using empirical equations here.

Reviewer #2 (Remarks on code availability):

#35. The code execution is reliable.

Response: Thank you for helping check the code!

Reviewer #3 (Remarks to the Author):

#36. Overall, this is a timely manuscript in which the authors sought to determine the net effect of atmospheric nitrogen deposition on rates of soil respiration in forests around the globe. They conducted a meta-analysis with data compiled through published papers cited in Web of Science and from the China National Knowledge Infrastructure Theses and Dissertations Database.

Here are my main constructive critiques of the paper:

Their dataset is biased toward Chinese collections given they looked at papers cited in Web of Science, but only dissertations from the “China National Knowledge Infrastructure Theses and Dissertations database”. It’s unclear why they used student papers from China and not elsewhere.

Response: We collected data from the Web of Science database because it covers the main-stream English-language journals. We also collected data from the China National Knowledge Infrastructure Theses and Dissertations Database because some Chinese students did not publish their data in English-language journals but only in their theses or dissertations. The first author of this study (who compiled the data) as a native Chinese speaker can read the masters' theses and doctoral dissertations written in Chinese. Therefore, we were able to take advantage of our bilingual skills and expand the size of our dataset. We believe that retrieving papers from the Web of Science and China National Knowledge Infrastructure Theses and Dissertations databases is a common practice in many global meta-analysis studies, such as Yan et al. (2022), Chen and Chen (2023), Liu et al. (2023). We added some text in L385–386 to clarify.

In the future, if we could have the help of collaborators with different language skills, we would also like to extend the data search to include published literature written in other languages, thereby further expanding our datasets.

#37. There are many places where the writing needs to be improved. For example, in the Highlights it says “Experimental and observational data agree on a new framework...” As written, it sounds like the data themselves are thinking and agreeing on something; the authors need to make it clearer that researchers can agree on a new framework, data cannot. This and other text throughout the paper should be revised in this way.

Response: Thank you for pointing this out. The entire manuscript has been revised to avoid using anthropomorphic statements.

#38. In general I suggest using fewer abbreviations. Many are not necessary. For example, why is “Pdec” used throughout and not just “N inputs”? If the authors want this paper to be read and understood by a broad audience, it would be much easier to read and follow if fewer abbreviations are used.

Response: Sorry for the inconvenience. We have reduced the use of abbreviations wherever possible. We also added a glossary of terms and abbreviations at the beginning of the paper (Box 1) to increase its readability.

#39. There are several places where the authors make a comparison, but don't finish it. See below for details.

Response: Thank you for pointing this out. We have revised accordingly.

The following detailed comments are meant to improve the manuscript:

Highlights

#40. Line 42: Change first bullet to be less anthropomorphic

Response: This sentence has been removed to comply with the formatting requirements. But we have revised the entire manuscript to avoid being anthropomorphic.

#41. Line 44: Change to “N deposition reduces...” and finish the statement “more frequently in N-saturated than non-N saturated”? Or “more frequently than “N-limited forests”? Or something else?

Response: This sentence has been removed to comply with the formatting requirements, but we have revised the entire manuscript to clearly state the comparisons.

#42. Line 46: “Responses of plant root and microbial respirations to N input differ in magnitude and pace”

- I suggest replacing this with the actual differences.

Response: This sentence has been removed to comply with the formatting requirements.

#43. Line 47: “Current level of N deposition enhanced global forest soil respiration by ~5%” relative to what? Pre-industrial levels of atmospheric deposition? Non-forest ecosystems?

Response: We intended to mean that 5% of the global forest soil respiration budget is attributable to atmospheric N deposition (i.e., global forest soil respiration budget would decrease by ~5% if there were no atmospheric N deposition). This sentence has been removed.

Abstract

#44. Lines 73-76: This text is unclear “Using a novel probabilistic approach, we estimated that N deposition decreased SR in 2.9% of global forests, mostly N-saturated forests in eastern China, western Europe, and eastern USA. But the net effect of N deposition increased the global forest SR budget by 5.1% (1.7 PgC yr⁻¹).” The title of the paper includes these three regions only (eastern China, western China, and eastern USA); what regions are the authors referring to about the “net effect of N deposition”?

Response: Here we intended to say that N deposition decreased SR in 2.9% of global forests. Those forests are mostly N-saturated forests, which are mainly located in eastern China, western Europe, and eastern USA.

Even in the same region, N deposition can increase and decrease SR depending on the background N availability in a forest. Therefore, in this study, we separately quantified the increased and decreased SR by N deposition (see Fig. 6 in the main text) and

reported here the net effect of N deposition (i.e., increased SR minus decreased SR).

This sentence has been revised to avoid being misleading (L49–51).

#45. Lines 73-75: Also related to this text: “Using a novel probabilistic approach, we estimated that N deposition decreased SR in 2.9% of global forests, mostly N-saturated forests in eastern China, western Europe, and eastern USA.”

-Are the authors saying that N deposition increased SR in the non N-saturated forests of eastern China, Western Europe, and eastern USA?

-Or, are they saying that N deposition increased SR in other regions of the world?

Here and elsewhere the authors often state the first half of a comparison, but don't complete it so it's difficult to assess the assertion they are making

Response: Sorry for being misleading. We intended to say that N deposition decreased SR in 2.9% of global forests. Those forests are mostly N-saturated forests, which are mainly located in eastern China, western Europe, and eastern USA. This sentence has been revised to clarify (L49–51). We also revised the entire manuscript to avoid using only the first half of a comparison.

#46. Lines 76-77: “If N pollution could be effectively controlled, global forest SR and its variability would decrease, thereby reducing the uncertainty in the projected terrestrial carbon dynamics.”

-It's unclear what evidence the authors are using to draw this final conclusion.

-I suggest they share results about variability and uncertainty before jumping from trends in N deposition effects on SR (Lines 73-76) to how reductions in N deposition would impact variability in SR and our predictions of this important process (Lines 76-77)

Response: Thanks for pointing it out. The result on SR variability was moved to the supplementary materials (Supp. Fig. S9) in a previous revision. As such, it may not be appropriate to appear in the abstract. This sentence has been revised to mention only SR but not its variability (L52–53).

#47. Line 70: Include the total number of experiments included in the data synthesis

Response: Revised accordingly (L46).

Introduction

#48. Line 80: Spell out “carbon dioxide” first time CO₂ is used

Response: Revised accordingly (L57).

#49. Lines 148-149: The authors need to make clear what the experimental unit is here. Why 1112 observations, but only 168 experiments? Did they keep data separate by tree species? Form of N addition? How did they handle time series data? Take averages over time before including in the data synthesis?

Response: In each experiment, observations from different plots (e.g., control, low N addition, high N addition plots) and different years are usually reported separately. These data were kept separate in our compiled dataset (CO₂_exp; downloadable from <https://data.mendeley.com/preview/wfdfjds6by?a=c296db55-569b-4c5f-8b65-d85e9e51dd3a>). In our analysis, to avoid the over-representation of data from the same experiment (i.e., data from the same plot-year may have been analyzed and reported in different papers), we summarized data from the same year and N addition level by taking their averages. Each site-year-N addition level was used as a data point when we calculated the response factors of SR to N addition. We have revised the Methods section to clarify (L416–423). This sentence was also revised (L121).

#50. Lines 156-159: Move this text to start the paragraph so the main goals are stated first, then followed by the methods (including # of observations and experiments included in the study)

Response: Thank you for your suggestion. We have revised accordingly (L117–120).

Figure 3:

#51. How were biomes defined? In other words, did the authors utilize type of biome noted by authors of published data? Did they compile latitude and longitude and categorize biomes themselves? Did they utilize mean annual precipitation and temperature?

Response: Different papers may use different forest biome classifications. For consistency, we used coordinates of each forest to extract the biome information from a global spatial dataset (Hansen et al. 2010). We have revised the Methods section to clarify (L410–411).

#52. It appears from Figure 3A-C that rates of total soil, root, and microbial respiration all decline with greater rates of experimental N inputs AND that the steepest declines occur at the lowest rates of N inputs (which is contrary to their hypotheses and conclusions). It is unclear why the authors describe these relationships as “increased SR by N input”

Response: Sorry to be misleading here. Fig. 3A–C and G–I panels show the response factor of increased SR to N input (f_a). For reasons specified in the response to Comment #3, it is very difficult to directly analyze the quantitative relationship between N input and the increased SR. In contrast, the relationship between f_a and N input rate is more

straightforward and quantifiable.

The response factor (f_a) is similar to the first-order derivative of the increased SR by N input. The negative, linear relationship between the response factor and N input rate provided us with a basis for quantifying the quadratic relationship between increased SR and N input (Fig. 3D–F, J–L). The quadratic curves shown in Fig. 3D–F, J–L supported our hypothesized biphasic relationship between N input and the increased SR.

We have revised Fig. 3 and legend to better show the logistic relationship between the panels.

#53. Figures 3A-C and G-I: I suggest showing the statistical results for both slopes (the red and gray lines in each)

Response: We hesitate to add additional text to the panels, because Fig. 3 may look too busy. Instead, we provided a supplemental table (Supp. Table S1) to clearly show the statistical results for all the piecewise models in Fig. 3 and Supp. Fig. S3.

#54. I understand the detailed Methods come below, but the authors need to explain in the legend for Figure 3 what the experimental unit is (each dot). Is it site? Tree species kept separate? How are time series from individual sites handled?

Response: In Fig. 3, each gray point represents a response factor (f_a , f_r) or soil respiration change (ΔSR_N) calculated using data from a pair of control and N addition plots from the same site-year. At the same site and year, observations from plots with the same N addition level were summarized by taking their averages. We have revised the legend for Fig. 3 (L839–840) and the Methods section (L415–422) to clarify.

Data from different years were treated separately to account for the potential influence of the duration of experimental N addition on SR responses. The tree species factor was not considered in this study. In most cases, N addition experiment was only conducted in one community (with one forest type) at each site. Thus, the current dataset can hardly support species-specific analyses.

#55. Line 241: The authors state “Under high N deposition, ΔSR_N decreased. The declining trends were more pronounced in tropical forests and N-saturated forests,…”

-However, it’s unclear how they are designating sites as “N-saturated” or not. It should be briefly defined here

Response: In a previous study (Cen et al. 2025), we determined the N limitation or saturation status of global forests using the sensitivity of soil N_2O emission to N deposition (s_N) as an indicator. In published papers, researchers reported the N limitation or saturation status of forest sites based on either the forest growth response to N addition, or the N leaching level. Those observations were used to determine the

s_N threshold between N-limited and N-saturated forests, based on which we further classified global forests to be N-limited or N-saturated.

In this study, using the derived N status map of global forests, we were able to separately analyze the SR responses to N addition in N-limited and N-saturated forests. We added some text here to clarify (L174–175).

#56. Lines 263-265: The authors state “Generally, P_{dec} increased with increasing N stress, which can result from high N inputs, forest N saturation, or long-term artificial N additions.”

-It’s unclear how the authors have defined “N stress” based on their results. It is clear conceptually that high N input sites that are N saturated can experience “stress”, but it’s unclear how the authors are utilizing their data to come to this conclusion.

-Is “stress” here defined as the “probability of forest soil respiration being decreased by N input”?

Response: Sorry for the unclarity here. “N stress” was intended to refer to a biochemically unfavorable condition for the fitness of organisms, which may result from high background N availability (or N saturation), high exogenous N inputs (deposition or fertilization), or long-term N additions. But because we have seen multiple comments on the use of “N stress”, this term has been deprecated in the revised manuscript.

This sentence has been revised as “Generally, P_{dec} increased with increasing N availability”.

#57. Line 303-304: The authors state “Globally, we found that current levels of N deposition increased SR in most forests, by a total of 1.8 ± 0.1 PgC yr⁻¹.”

-It would be stronger for the authors to report this in terms of kg N added through atmospheric inputs.

Response: This is a good suggestion. We have added some text here to report “a net effect size of +84 kgC kgN⁻¹”.

Additionally, we provided a Supp. Table S5 to show the different SR response factors of forests in different biomes (tropical/temperate/boreal) and different N status (N-limited/N-saturated).

#58. Lines 303-304 and Table 1: The results would be more intuitive and powerful if the authors expressed the reduction or increase in soil respiration caused by N inputs as something like: “PgC kg N inputs⁻¹ yr⁻¹”

Response: This is a good suggestion. Although we hesitate to add columns to Table 1 for aesthetic reasons (the revised table would be too wide to display on one page), we

have provided a Supp. Table S5 to supplement Table 1 and show the different SR response factors (in units of $\text{kgCO}_2 \text{ kgN}^{-1}$) for forests in different biomes and different N status.

Discussion (no line numbers provided by authors)

Methods Section (no line numbers provided by authors)

#59. Section 4.1: The authors need to explain how they handled multiple samplings from the same study (1112 observations). Did they take the average across time so only one soil respiration rate for control and N-addition plots were included? Did they keep growing season from winter measurements separate or did they combine them?

Response: In our analysis, to avoid the over-representation of data from the same experiment (i.e., data from the same plot-year may have been analyzed and reported in different papers), we summarized data from the same year and N addition level by taking their averages. Each unique site-year-N addition level combination was used as a data point when we calculated the response factors of SR to N addition.

For instance, if one experimental site has nine plots receiving three different N addition treatments (including the control; each treatment has three replications) and the experiment was conducted and reported for three years, we will have 27 annual-scale observations from that site. Observations from the same year and N addition level will be summarized by taking the averages, meaning that data from the 9 unique year-N addition level combinations will be used for further analyses such as the calculation of response factors. We have revised the Methods section to clarify (L415–422).

Only annual soil respiration rates were collected from published papers. We did not separately collect data for the growing and non-growing seasons. This study focused only on the effects of N deposition on annual soil respiration in forests. Interested researchers could investigate the effects of N deposition on the seasonality of soil respiration using a dataset with high temporal resolution.

Reviewer #3 (Remarks on code availability):

Bibw

Thanks again.

Yours sincerely,

Nianpeng He

References

- Aber, J. D., W. McDowell, K. Nadelhoffer, A. Magill, G. Berntson, M. Kamakea, S. McNulty, W. Currie, L. Rustad, and I. Fernandez. 1998. Nitrogen saturation in temperate forest ecosystems: hypotheses revisited. *Bioscience* **48**:921-934.
- Bae, K., T. J. Fahey, R. D. Yanai, and M. Fisk. 2015. Soil Nitrogen Availability Affects Belowground Carbon Allocation and Soil Respiration in Northern Hardwood Forests of New Hampshire. *Ecosystems* **18**:1179-1191.
- Bobbink, R., K. Hicks, J. Galloway, T. Spranger, R. Alkemade, M. Ashmore, M. Bustamante, S. Cinderby, E. Davidson, F. Dentener, B. Emmett, J. W. Erisman, M. Fenn, F. Gilliam, A. Nordin, L. Pardo, and W. De Vries. 2010. Global assessment of nitrogen deposition effects on terrestrial plant diversity: a synthesis. *Ecological Applications* **20**:30-59.
- Bond-Lamberty, B., and A. Thomson. 2010. Temperature-associated increases in the global soil respiration record. *Nature* **464**:579-U132.
- Brown, J. H., J. F. Gillooly, A. P. Allen, V. M. Savage, and G. B. West. 2004. TOWARD A METABOLIC THEORY OF ECOLOGY. *Ecology* **85**:1771-1789.
- Cen, X., N. He, K. Van Sundert, C. Terrer, K. Yu, M. Li, L. Xu, L. He, and K. Butterbach-Bahl. 2025. Global patterns of nitrogen saturation in forests. *One Earth* **8**:101132.
- Chen, C., and H. Y. H. Chen. 2023. Mapping global nitrogen deposition impacts on soil respiration. *Science of the Total Environment* **871**:161986.
- Chen, S., J. Zou, Z. Hu, H. Chen, and Y. Lu. 2014. Global annual soil respiration in relation to climate, soil properties and vegetation characteristics: Summary of available data. *Agricultural and Forest Meteorology* **198-199**:335-346.
- de Vries, W., E. Z. Du, and K. Butterbach-Bahl. 2014. Short and long-term impacts of nitrogen deposition on carbon sequestration by forest ecosystems. *Current Opinion in Environmental Sustainability* **9-10**:90-104.
- Egidi, E., C. Coleine, M. Delgado-Baquerizo, and B. K. Singh. 2023. Assessing critical thresholds in terrestrial microbiomes. *Nature Microbiology* **8**:2230-2233.
- Forsmark, B., A. Nordin, N. I. Maaroufi, T. Lundmark, and M. J. Gundale. 2020. Low and High Nitrogen Deposition Rates in Northern Coniferous Forests Have Different Impacts on Aboveground Litter Production, Soil Respiration, and Soil Carbon Stocks. *Ecosystems* **23**:1423-1436.
- Geng, J., S. L. Cheng, H. J. Fang, G. R. Yu, X. Y. Li, G. Y. Si, S. He, and G. X. Yu. 2017. Soil nitrate accumulation explains the nonlinear responses of soil CO₂ and CH₄ fluxes to nitrogen addition in a temperate needle-broadleaved mixed forest. *Ecological Indicators* **79**:28-36.
- Gilliam, F. S. 2006. Response of the herbaceous layer of forest ecosystems to excess nitrogen deposition. *Journal of Ecology* **94**:1176-1191.
- Gleser, L. J., and I. Olkin. 2009. Stochastically dependent effect sizes. *The Handbook of Research Synthesis and Meta-analysis*:357-376.
- Hansen, M. C., S. V. Stehman, and P. V. Potapov. 2010. Quantification of global gross forest cover loss. *Proceedings of the National Academy of Sciences* **107**:8650-8655.
- Huang, N., L. Wang, X. P. Song, T. A. Black, R. S. Jassal, R. B. Myneni, C. Y. Wu, L. Wang, W. J. Song, D. B. Ji, S. S. Yu, and Z. Niu. 2020. Spatial and temporal variations in global soil respiration and their relationships with climate and land cover. *Science Advances* **6**:11.
- Janssens, I. A., W. Dieleman, S. Luysaert, J. A. Subke, M. Reichstein, R. Ceulemans, P. Ciais, A. J. Dolman, J. Grace, G. Matteucci, D. Papale, S. L. Piao, E. D. Schulze, J. Tang, and B. E. Law. 2010. Reduction of forest soil respiration in response to nitrogen deposition. *Nature Geoscience* **3**:315-322.
- Li, Q., X. Z. Song, S. X. Chang, C. H. Peng, W. F. Xiao, J. B. Zhang, W. H. Xiang, Y. Li, and W. F. Wang. 2019. Nitrogen depositions increase soil respiration and decrease temperature sensitivity in a Moso bamboo forest. *Agricultural and Forest Meteorology* **268**:48-54.
- Liu, Y., M. Men, Z. P. Peng, H. Y. H. Chen, Y. H. Yang, and Y. F. Peng. 2023. Spatially

- explicit estimate of nitrogen effects on soil respiration across the globe. *Global Change Biology* **29**:3591-3600.
- Mason, R. E., J. M. Craine, N. K. Lany, M. Jonard, S. V. Ollinger, P. M. Groffman, R. W. Fulweiler, J. Angerer, Q. D. Read, P. B. Reich, P. H. Templer, and A. J. Elmore. 2022. Evidence, causes, and consequences of declining nitrogen availability in terrestrial ecosystems. *Science* **376**:261-+.
- Michaelis, L., and M. L. Menten. 1913. Die kinetik der invertinwirkung. *Biochem. z* **49**:352.
- Noble, D. W. A., M. Lagisz, R. E. O'Dea, and S. Nakagawa. 2017. Nonindependence and sensitivity analyses in ecological and evolutionary meta-analyses. *Molecular Ecology* **26**:2410-2425.
- Peng, B., J. F. Sun, J. Liu, Z. W. Xia, and W. W. Dai. 2021. Relative contributions of different substrates to soil N₂O emission and their responses to N addition in a temperate forest. *Science of the Total Environment* **767**:8.
- Pichler, M., and F. Hartig. 2023. Machine learning and deep learning—A review for ecologists. *Methods in Ecology and Evolution* **14**:994-1016.
- Raich, J. W., and W. H. Schlesinger. 1992. The global carbon dioxide flux in soil respiration and its relationship to vegetation and climate. *Tellus B* **44**:81-99.
- Reay, D. S., F. Dentener, P. Smith, J. Grace, and R. A. Feely. 2008. Global nitrogen deposition and carbon sinks. *Nature Geoscience* **1**:430-437.
- Reich, P. B., M. G. Tjoelker, J.-L. Machado, and J. Oleksyn. 2006. Universal scaling of respiratory metabolism, size and nitrogen in plants. *Nature* **439**:457-461.
- Ritz, C., F. Baty, J. C. Streibig, and D. Gerhard. 2016. Dose-response analysis using R. *Plos One* **10**:e0146021.
- Treseder, K. K. 2008. Nitrogen additions and microbial biomass: a meta-analysis of ecosystem studies. *Ecology Letters* **11**:1111-1120.
- Wang, C., F. Ren, X. Zhou, W. Ma, C. Liang, J. Wang, J. Cheng, H. Zhou, and J.-S. He. 2020. Variations in the nitrogen saturation threshold of soil respiration in grassland ecosystems. *Biogeochemistry* **148**:311-324.
- Wang, C., and Y. Tang. 2019. Responses of plant phenology to nitrogen addition: a meta-analysis. *Oikos* **128**:1243-1253.
- Wang, W., W. Chen, and S. Wang. 2010. Forest soil respiration and its heterotrophic and autotrophic components: Global patterns and responses to temperature and precipitation. *Soil Biology and Biochemistry* **42**:1236-1244.
- Wang, W. J., R. C. Dalal, P. W. Moody, and C. J. Smith. 2003. Relationships of soil respiration to microbial biomass, substrate availability and clay content. *Soil Biology and Biochemistry* **35**:273-284.
- Wang, Z., A. Xing, and H. Shen. 2023. Effects of nitrogen addition on the combined global warming potential of three major soil greenhouse gases: A global meta-analysis. *Environmental Pollution* **334**:121848.
- Xing, A. J., E. Z. Du, H. H. Shen, L. C. Xu, W. de Vries, M. Y. Zhao, X. Y. Liu, and J. Y. Fang. 2022a. Nonlinear responses of ecosystem carbon fluxes to nitrogen deposition in an old-growth boreal forest. *Ecology Letters* **25**:77-88.
- Xing, A. J., E. Z. Du, H. H. Shen, L. C. Xu, M. Y. Zhao, X. Y. Liu, and J. Y. Fang. 2022b. High-level nitrogen additions accelerate soil respiration reduction over time in a boreal forest. *Ecology Letters* **25**:1869-1878.
- Xu, C., X. Xu, C. Ju, H. Y. H. Chen, B. J. Wilsey, Y. Luo, and W. Fan. 2021. Long-term, amplified responses of soil organic carbon to nitrogen addition worldwide. *Global Change Biology* **27**:1170-1180.
- Yan, W., Y. Zhong, J. Yang, Z. Shangguan, and M. S. Torn. 2022. Response of soil greenhouse gas fluxes to warming: A global meta-analysis of field studies. *Geoderma* **419**:115865.
- Yu, H. 2019. Effects of changing soil acidity/alkalinity on soil nitrogen and greenhouse gas fluxes in a *Pinus tabulaeformis* forest in Taiyue. Master's thesis. Beijing Forestry University.
- Yu, H., Y. Chen, H. Zhang, and Z. Zhou. 2019. The effect of inorganic nitrogen addition on soil nitrogen and greenhouse gas flux for the *Pinus tabulaeformis* forest in Taiyue

Mountain, Shanxi Province. *JOURNAL OF NANJING FORESTRY UNIVERSITY* **62**:85.

Zhao, B. 2019. Soil Respiration in Response to Thinning and Simulated Nitrogen Deposition in *Pinus Tabuliformis* Forest in Taiyue Mountain. Doctoral dissertation. Beijing Forestry University.

Zhao, Z., X. Ding, G. Wang, and Y. Li. 2024. 30 m Resolution Global Maps of Forest Soil Respiration and Its Changes From 2000 to 2020. *Earth's Future* **12**:e2023EF004007.

Zheng, M., T. Zhang, Y. Luo, J. Liu, X. Lu, Q. Ye, S. Wang, J. Huang, Q. Mao, J. Mo, and W. Zhang. 2022. Temporal patterns of soil carbon emission in tropical forests under long-term nitrogen deposition. *Nature Geoscience* **15**:1002-1010.

Point-to-point responses to the comments of reviewers

"A general framework for nitrogen deposition effects on soil respiration in global forests" [NCOMMS-25-10077A]

Dear reviewer,

Thank you for your comments. To address the remaining major concern of the reviewer, we have substantially revised the relevant sections of the manuscript, including the Introduction, Results, Discussion, and Methods sections of the manuscript, as well as the main figures (Figs. 1–3). As you suggested, the revised manuscript now contains an integrated analysis of the positive and negative responses of soil respiration to nitrogen deposition (see L175–179, 529–543; Supplementary Text S2, Fig. S6). This analysis revealed the same pattern as our previous separate analyses of positive and negative responses (Fig. 7). The data and R code for the analysis are available at a public repository (<https://data.mendeley.com/preview/wdfjds6by?a=c296db55-569b-4c5f-8b65-d85e9e51dd3a>).

Here, we are providing three files to show how we revised the manuscript: 1) revised manuscript with tracked changes; 2) revised supplementary materials; 3) point-by-point response to the reviewers' comments (attached below).

Thank you again for your time and suggestions, which helped us improve our manuscript. We would appreciate it if you could consider our manuscript again for publication.

Yours sincerely,

Nianpeng He

Institute of Geographic Sciences and Natural Resources Research, CAS

Datun Road A 11, Chaoyang District, Beijing 100101, P.R. China

E-mail: henp@igsnrr.ac.cn

Tel: 86-10-64889263

Fax: 86-10-64889432

Response to reviewer's comments:

Reviewers' comments:

Reviewer #1 (Remarks to the Author):

#1. I appreciate the authors' detailed responses and their efforts to clarify the rationale behind their analytical approach. However, I remain unconvinced by the decision to separate the analysis of positive and negative effects on SR, as no compelling theoretical or empirical justification has been provided to support this separation. Since this constitutes a fundamental step in their analytical framework, it casts doubt on the validity of the subsequent results and conclusions.

In their response, the authors suggest that increased SR results from "N-facilitated biomass growth," while decreased SR arises from "N-induced biodiversity loss." However, numerous grassland experiments have demonstrated that both biomass increase and biodiversity loss can occur simultaneously under the same nitrogen treatments (e.g., Bai et al., 2009, *Global Change Biology*, <https://doi.org/10.1111/j.1365-2486.2009.01950.x>). This evidence supports the point I raised earlier: both positive and negative mechanisms may operate concurrently within the same plot, making it problematic to analyze them in isolation.

Response: Thank you. Indeed, N could facilitate biomass growth and increase metabolic respiration (Fog 1988, Brown et al. 2004). Also, excessive N (and the disruption of biotic stoichiometry and soil pH) could cause the N-sensitive species to be replaced by N-tolerant species (Huang et al. 2015, Hedwall et al. 2017), leading to a loss of species and a decrease in respiration, at least temporarily, during the transition phase (Bobbink et al. 2010, Zheng et al. 2022). We agree that these two mechanisms could both occur under the same N treatments and at the same sites. However, in a specific plot, it is more likely that the two mechanisms function sequentially (i.e., N additions within a favorable range facilitate biomass growth, but accumulated exogenous N over time becomes detrimental and leads to the loss of N-sensitive species [H1]) than interactively (i.e., N addition facilitates biomass growth, which leads to the gradual, competitive exclusion of N-sensitive species in the meantime [H2]). Band et al. conducted a global meta-analysis using experimental N addition data from grasslands. They tested both hypotheses and found that the detrimental effect of N (i.e., H1) is the major driver of species loss in N addition experiments conducted in grasslands, rather than the interactive effect of biomass growth (i.e., H2)(Band et al. 2022). The temporal isolation of the two mechanisms suggests the possibility of separately analyzing the increased SR (likely dominated by N-facilitated biomass growth) and the decreased SR (likely dominated by N-induced species loss).

Nevertheless, many studies have observed both increased biomass and decreased species richness in the same grassland N addition site-years (e.g., Bai et al. (2010)). One possible explanation is that, annual changes in biomass (and productivity estimated by the harvesting method) reflect the average production over the year, which do not

sufficiently demonstrate the temporal changes of community composition and function. Temporary species loss and decreased respiration may be obscured by intra-annual species gain (Zhao et al. 2022) and biomass increase following community rebuilding. The rapid replacement of N-sensitive herbaceous plants with N-tolerant ones may explain why plant biomass in grasslands usually responds positively to N enrichment, while other ecosystems with more complex plant forms (e.g., forests) may witness both positive and negative responses (De Schrijver et al. 2011).

In addition to the mechanistic validity of analyzing the isolated increases and decreases of SR over time, a realistic reason why we had to analyze the increased and decreased SR separately is that the different characteristics of the mechanisms require different mathematical models (fig. 1; Fig. 1 in the main text). The N-induced biomass growth is a continuous process. In contrast, the N-induced species loss is often abrupt due to the resilience of the community to environmental (nitrogen) stress (Zheng et al. 2022, Egidi et al. 2023). Considering the different characteristics of the two mechanisms underlying the effect of N input on soil respiration (i.e., continuous vs. discrete), we had to separate the positive response (i.e., a net increase of SR in the N addition plot as compared to the control plot, which may be attributable to the dominant effect of continuous biomass growth; fig. 1) and the negative response (i.e., a net decrease of SR in the N addition plot as compared to the control plot, which may be attributable to the dominant effect of abrupt species loss), so that we could apply different mathematical models to quantify the SR responses.

fig. 1 Potential mechanisms underlying changes in soil respiration due to increasing nitrogen availability. This figure has been incorporated into the revised Fig. 1 in the main text.

On the other hand, the results of our analysis agrees with that of Bai et al. (2010) in that the same level of N input could result in positive or negative responses (although in different forests). This is precisely why we had to define and quantify the “probability

of SR being decreased by N input” (Pdec; Fig. 5) before integrating the two models for positive/negative SR responses and evaluating the overall effect of N deposition on global forest SR.

We have revised the entire manuscript to clarify the dominant mechanisms underlying the observed increases/decreases in SR (L93–121; Fig. 1 in the main text). We cited Bai et al. (2010) to support the effects of N on biomass and species richness (see L93–94). Analyzing the increase or decrease of SR separately allowed us to apply different mathematical models to quantify SR changes (see L138–141 for the revisions). We also incorporated fig. 1 to produce a new Fig. 1 in the main text, which further clarifies the mechanistic basis for distinguishing between positive and negative responses.

To address your concerns and provide additional empirical support for our conclusions, this revision highlights an integrated analysis that does not distinguish between positive and negative SR responses (see L175–179, 529–543; Supplementary Text S2, Fig. S6). We have previously used this method and integrated response factor to assess soil CH₄ flux responses to N input (Cen et al. 2024). Specifically, we calculated the integrated response factors using SR observations from N limited forests in which N was added experimentally (*CO₂_exp_NL* dataset in the main text). Forest that were initially N-saturated forests were not included in this analysis because only the N-limited forests would undergo the entire process (including the community collapse and reconstruction phase) and exhibit a bimodal SR pattern under increasing N inputs according to our proposed framework (Fig. 7). Integrated response factor (f_{int}) was calculated as

$$f_{int} = \frac{(SR_2 - SR_1)}{(N_2 - N_1)}$$

where SR_1 and SR_2 (unit: kgCO₂ ha⁻¹ yr⁻¹) were the soil respiration rates observed under two different N input rates of N_1 and N_2 (kgN ha⁻¹ yr⁻¹), respectively. To make the f_{int} comparable across forest sites, f_{int} was calculated within four defined N input ranges, ≤ 50 kgN ha⁻¹ yr⁻¹, 50–100 kgN ha⁻¹ yr⁻¹, 100–150 kgN ha⁻¹ yr⁻¹, and ≥ 150 kgN ha⁻¹ yr⁻¹.

We found that with increasing N input, the integrated response factor (f_{int}) of SR to N input changed from positive to negative and then to positive across all biomes (Supplementary Fig. S6). This result suggests that in N-limited forests, SR first increases, then decreases, and then increases in response to increased N input. These findings align with those from our separate analysis of positive and negative responses and support our proposed framework (i.e., Fig. 7 in the main text).

Fig. S6 Integrated response factors of soil respiration rate to N input (f_{int}) were calculated for four different N input levels in N-limited experimental forests. Columns indicate the mean f_{int} corresponding to each N addition level in each biome; error bars indicate the standard errors of the mean values. Numbers on the top of the panel are the total number of calculated f_{int} values corresponding to each N addition level in each biome.

#2. Also, the authors mentioned that the positive N effect on SR is biphasic because of “high N becomes toxic”. I would argue that “high N becomes toxic” is also one of the reasons for negative responses of SR to N addition. This again challenges the validity of separating these effects analytically.

Response: We agree that excessive N could lead to inhibited biomass growth and also species loss (Fig. 1 in the main text; L111–112). We conducted separate analyses of the positive and negative responses to apply different mathematical models and quantify the effects of N deposition on SR. A positive response is determined by a net increase in SR in the N addition plot compared to the control plot, suggesting that the N-induced, continuous biomass growth is dominant. The negative response is determined by the net decrease of SR in the N addition plot compared to the control plot, suggesting that N-induced, abrupt species loss is dominant. We revised the manuscript to clarify the potential mechanisms underlying these responses (L93–121, 138–141).

#3. Overall, I find the separation of positive and negative SR responses to nitrogen addition conceptually confusing and methodologically unsubstantiated. Without stronger evidence or theoretical support for this approach, I cannot endorse the results or conclusions that depend on it.

Response: We have thoroughly revised the manuscript to clarify that the separate analyses of the positive and negative responses were due to the different dominant mechanisms underlying the positive response (likely dominated by continuous biomass

change) and the negative response (likely dominated by abrupt species loss). This separation enabled us to develop a continuous model (i.e., biphasic model shown in Fig. 3 in the main text) for quantifying the positive response of SR to any N input level, and an abrupt transition model (see Fig. 3 in the main text) for quantifying the negative response of SR. By quantifying another key variable, probability of SR being decreased by N input (P_{dec} ; Fig. 5), we were able to integrate the two models for positive/negative SR responses and evaluate the overall effect of any N deposition level on global forest SR (fig. 2).

fig. 2 Illustration and explanation of the workflow how we handled the experimental N addition data. Experimental N addition data were separated to build mathematical models for positive and negative responses of soil respiration (SR) to N inputs. On the basis, we quantified the N deposition-induced increase and decrease of SR. Because the same N input rate can increase or decrease SR, we quantified a “probability of soil respiration being decreased by N input” (P_{dec}) for integrating the positive and negative effects and reveal the overall effect of N deposition on SR. SR_{CL} : soil respiration rate measured in the control plot receiving natural nitrogen deposition (N_{dep}). SR_{+N} : soil respiration rate measured in the experimental plot receiving an additional N input (N_{add}). ΔSR_{N-inc} : increased SR by N input. ΔSR_{N-dec} : decreased SR by N input. f_{pos} : positive response factor of increased SR to N input. f_{neg} : negative response factor of decreased SR to N input (see Supplementary Text S1 for the response factors derived from biochemical and ecological theories). P_{dec} : Probability of SR being decreased by N input, estimated as the proportion of experimental N addition forests where a N input rate caused a decrease in soil respiration. This figure has been incorporated into the revised Fig. 2 in the main text.

Reviewer #2 (Remarks to the Author):

#4. The authors have made thorough and appropriate revisions in response to the Editor's and reviewers' previous comments. I am satisfied with the current version of the manuscript and have no further comments.

Response: Thank you!

Reviewer #2 (Remarks on code availability):

#5. The code execution is reliable.

Response: Thanks for helping check the code execution.

Thanks again.

Yours sincerely,

Nianpeng He

References

- Bai, Y., J. Wu, C. M. Clark, S. Naeem, Q. Pan, J. Huang, L. Zhang, and X. Han. 2010. Tradeoffs and thresholds in the effects of nitrogen addition on biodiversity and ecosystem functioning: evidence from inner Mongolia Grasslands. *Global Change Biology* **16**:358-372.
- Band, N., R. Kadmon, M. Mandel, and N. DeMalach. 2022. Assessing the roles of nitrogen, biomass, and niche dimensionality as drivers of species loss in grassland communities. *Proceedings of the National Academy of Sciences* **119**:e2112010119.
- Bobbink, R., K. Hicks, J. Galloway, T. Spranger, R. Alkemade, M. Ashmore, M. Bustamante, S. Cinderby, E. Davidson, F. Dentener, B. Emmett, J. W. Erisman, M. Fenn, F. Gilliam, A. Nordin, L. Pardo, and W. De Vries. 2010. Global assessment of nitrogen deposition effects on terrestrial plant diversity: a synthesis. *Ecological Applications* **20**:30-59.
- Brown, J. H., J. F. Gillooly, A. P. Allen, V. M. Savage, and G. B. West. 2004. TOWARD A METABOLIC THEORY OF ECOLOGY. *Ecology* **85**:1771-1789.
- Cen, X., N. He, M. Li, L. Xu, X. Yu, W. Cai, X. Li, and K. Butterbach-Bahl. 2024. Suppression of Nitrogen Deposition on Global Forest Soil CH₄ Uptake Depends on Nitrogen Status. *Global Biogeochemical Cycles* **38**:e2024GB008098.
- De Schrijver, A., P. De Frenne, E. Ampoorter, L. Van Nevel, A. Demey, K. Wuyts, and K. Verheyen. 2011. Cumulative nitrogen input drives species loss in terrestrial ecosystems. *Global Ecology and Biogeography* **20**:803-816.
- Egidi, E., C. Coleine, M. Delgado-Baquerizo, and B. K. Singh. 2023. Assessing critical thresholds in terrestrial microbiomes. *Nature Microbiology* **8**:2230-2233.
- Fog, K. 1988. THE EFFECT OF ADDED NITROGEN ON THE RATE OF DECOMPOSITION OF ORGANIC MATTER. *Biological Reviews* **63**:433-462.
- Hedwall, P. O., J. Bergh, and J. Brunet. 2017. Phosphorus and nitrogen co-limitation of forest ground vegetation under elevated anthropogenic nitrogen deposition. *Oecologia* **185**:317-326.
- Huang, Y. M., R. H. Kang, J. Mulder, T. Zhang, and L. Duan. 2015. Nitrogen saturation, soil acidification, and ecological effects in a subtropical pine forest on acid soil in southwest China. *Journal of Geophysical Research-Biogeosciences* **120**:2457-2472.
- Zhao, M., H. Zhang, C. C. Baskin, C. Wei, J. Yang, Y. Zhang, Y. Jiang, L. Jiang, and X. Han. 2022. Intra-annual species gain overrides species loss in determining species richness in a typical steppe ecosystem after a decade of nitrogen enrichment. *Journal of Ecology* **110**:1942-1956.
- Zheng, M., T. Zhang, Y. Luo, J. Liu, X. Lu, Q. Ye, S. Wang, J. Huang, Q. Mao, J. Mo, and W. Zhang. 2022. Temporal patterns of soil carbon emission in tropical forests under long-term nitrogen deposition. *Nature Geoscience* **15**:1002-1010.